# Dirichlet Mechanism for Differentially Private KL Divergence Minimization

**Donlapark Ponnoprat**                                                *donlapark.p@cmu.ac.th*
*Department of Statistics*
*Chiang Mai University*

**Reviewed on OpenReview:** *https://openreview.net/forum?id=lmr2WwlaFc*

## Abstract

Given an empirical distribution $f(x)$ of sensitive data $x$, we consider the task of minimizing $F(y) = D_{\mathrm{KL}}(f(x)\|y)$ over a probability simplex, while protecting the privacy of $x$. We observe that, if we take the exponential mechanism and use the KL divergence as the loss function, then the resulting algorithm is the *Dirichlet mechanism* that outputs a single draw from a Dirichlet distribution. Motivated by this, we propose a Rényi differentially private (RDP) algorithm that employs the Dirichlet mechanism to solve the KL divergence minimization task. In addition, given $f(x)$ as above and $\hat{y}$ an output of the Dirichlet mechanism, we prove a probability tail bound on $D_{\mathrm{KL}}(f(x)\|\hat{y})$, which is then used to derive a lower bound for the sample complexity of our RDP algorithm. Experiments on real-world datasets demonstrate advantages of our algorithm over Gaussian and Laplace mechanisms in supervised classification and maximum likelihood estimation.

## 1 Introduction

KL divergence is the most commonly used divergence measure in probabilistic and Bayesian modeling. In a probabilistic model, for example, we estimate the model's parameters by maximizing the likelihood function of the parameters, which in turn is equivalent to minimizing the KL divergence between the empirical distribution and the model's distribution. In supervised classification, a standard way to fit a classifier is by minimizing the cross-entropy of the model's predictive probabilities, which is equivalent to minimizing the KL divergence between the class-conditional empirical distribution and the model's predictive distribution.

Such models are widely used in medical fields, social sciences and businesses, where they are used to analyze sensitive personal information. Without privacy considerations, releasing a model to public might put the personal data at risk of being exposed to privacy attacks, such as membership inference attacks (Shokri et al., 2017; Ye et al., 2022). To address the model's privacy issue, we should focus on its building blocks: the KL divergences. How can we minimize the KL divergence over the model's parameters, while keeping the data private?

Differential Privacy (Dwork et al., 2006a;b) provides a framework for quantitative privacy analysis of algorithms that run on sensitive personal data. Under this framework, one aims to design a task-specific algorithm that preserves the privacy of the inputs, while keeping the "distance" between the privatized output and the true output sufficiently small. A simple and well-studied technique is to add a small random noise sampled from a zero-centered probability distribution, such as the Laplace and Gaussian distributions. Another technique is to sample an output from a distribution, with greater probabilities of obtaining points that are closer to the true output, such as the exponential mechanism (McSherry & Talwar, 2007). These techniques have been deployed in many privacy-preserving tasks, from simple tasks such as private counting and histogram queries (Dwork et al., 2006a;b) to complex tasks such as deep learning (Abadi et al., 2016).

In this work, we are interested in a setting where our algorithm outputs an empirical distribution $f(x)$ of some sensitive data $x$. To protect the privacy of individuals in $x$, we keep $f(x)$ hidden, and instead release another discrete distribution that approximates $f(x)$ in KL divergence. This setting may not arise, for example, in

the task of releasing a normalized histogram, as the distance between histograms is often measured in $\ell^1$ or $\ell^2$. Nonetheless, there are many tasks where KL divergence arises naturally. Prominent examples are those in probabilistic modeling, where the outputs—the model's estimated parameters—are obtained from likelihood maximization. Another examples are those in Bayesian modeling, where models are evaluated with adjusted negative log-likelihood scores, such as the Akaike information criterion (AIC) and Bayesian information criterion (BIC). It is also increasingly common in Bayesian practice to evaluate the model with out-of-sample log-likelihood (Vehtari et al., 2016). Again, minimizing these criteria can be formulated as minimizing the KL divergence.

A simple approach to privatize a discrete probability distribution is by adding some random noises from a probability distribution. However, the KL divergence does not behave smoothly with the additive noises, as the following example illustrates: consider count data of 10 people, interpreted as a normalized histogram: $p = (0.1, 0.9)$. Suppose that we draw two sets of noises $z_1 = (-0.090, 0.045)$ and $z_2 = (-0.099, -0.038)$ from Laplace(1/10). Here, the $\ell^1$-distances between $p$ and its noisy versions are 0.135 and 0.137, a very small difference. On the other hand, the KL divergences between between $p$ and its noisy versions are 0.186 and 0.499, a 2.68 times increase. This example illustrates that adding noises to a discrete probability vector, even at a small scale, could result in a noisy vector that is too far away from the original vector in terms of KL divergence.

We instead consider the exponential mechanism, a differentially private algorithm that approximately minimizes user-defined *loss functions*. It turns out that, by taking the loss function to be the KL divergence, the exponential mechanism turns into one-time sampling from a Dirichlet distribution; we shall call this the *Dirichlet mechanism.*

The Dirichlet mechanism, however, does not inherit the differential privacy guarantee of the exponential mechanism: the guarantee in (McSherry & Talwar, 2007) requires the loss function to be bounded above, while the KL divergence can be arbitrarily large. In fact, using the original definition of differential privacy (Dwork et al., 2006b), the Dirichlet mechanism is not differentially private (see Appendix A). We thus turn to a relaxation of differential privacy. Specifically, using the notion of the Rényi differential privacy (Mironov, 2017), we study the Dirichlet mechanism and its utility in terms of KL divergence minimization.

## 1.1 Overview of Our results

Below are summaries of our results.

**§3  Privacy.** We propose a version of the Dirichlet mechanism (Algorithm 1) that satisfies the Rényi differential privacy (RDP). In this algorithm, we need to evaluate a polygamma function and find the root of a strictly increasing function. Our algorithm is easy to implement, as polygamma functions, root-finding methods and Dirichlet distributions are readily available in many scientific programming languages.

**§4  Utility.** We derive a probability tail bound for $D_{\mathrm{KL}}(p\|q)$ when $q$ is drawn from a Dirichlet distribution (Theorem 2). From this, we derive a lower bound for the sample complexity of Algorithm 1 that attains a target privacy guarantee, both in general case and on categorical data.

**§5  Experiments.** We compare the Dirichlet mechanism against the Gaussian and Laplace mechanisms for two learning tasks: naïve Bayes classification and maximum likelihood estimation of Bayesian networks—both tasks can be done with KL divergence minimization. Experiments on real-world datasets show that the Dirichlet mechanism provides smaller cross-entropy loss in classification, and larger log-likelihood in parameter estimation, than the other mechanisms at the same level of privacy guarantee.

## 1.2 Notations

In this paper, all vectors are $d$-dimensional, where $d \geq 2$. The number of observations is always $N$. Let $[d] := [1, \ldots, d]$. For any $u \in \mathbb{R}^d$, we let $u_i$ be the $i$-th coordinate of $u$, and for any vector-valued function $f : \mathcal{X} \to \mathbb{R}^d$, we let $f_i$ be that $i$-th component of $f$. Let $\mathbb{R}^d_{\geq 0}$ be the set of $d$-tuples of non-negative real numbers, and $\mathbb{R}^d_{>0}$ be the set of $d$-tuples of positive real numbers. Denote the probability simplex by

$$S^{d-1} := \left\{ p \in \mathbb{R}^d_{\geq 0} : \sum_i p_i = 1 \right\}.$$

For any $u, u' \in \mathbb{R}^d$ and scalar $r > 0$, we write $u + u' := (u_1 + u'_1, \ldots, u_d + u'_d)$ and $ru := (ru_1, \ldots, ru_d)$. For any positive-valued functions $f, f'$, the notation $f(x) \propto f'(x)$ means $f(x) = Cf'(x)$ for some constant $C > 0$ and $f(x) \approx f'(x)$ means $cf'(x) \leq f(x) \leq Cf'(x)$ for some $c, C > 0$. Lastly, $\|u\|_2 := \sqrt{u_1^2 + \ldots + u_d^2}$ is the $\ell^2$ norm of $u$ and $\|u\|_\infty := \max_i |u_i|$ is the $\ell^\infty$ norm of $u$.

## 2 Background and related work

### 2.1 Privacy models

We say that two datasets are *neighboring* if they differ on a single entry. Here, an *entry* can be a row of the datasets, or a single attribute of a row.

**Definition 2.1** (Pure and Approximate differential privacy (Dwork et al., 2006a;b)). A randomized mechanism $M : \mathcal{X}^n \to \mathcal{Y}$ is $(\varepsilon, \delta)$-differentially private ($(\varepsilon, \delta)$-DP) if for any two neighboring datasets $x$ and $x'$ and all events $E \subset \mathcal{Y}$,

$$\Pr[M(x) \in E] \leq e^\varepsilon \Pr[M(x') \in E] + \delta. \tag{1}$$

If $M$ is $(\varepsilon, 0)$-DP, then we say that it is $\varepsilon$-differential private ($\varepsilon$-DP).

The term *pure differential privacy* (pure DP) refers to $\varepsilon$-differential privacy, while *approximate differential privacy* (approximate DP) refers to $(\varepsilon, \delta)$-DP when $\delta > 0$.

In this paper, we shall concern ourselves with Rényi differential privacy, a relaxed notion of differential privacy defined in terms of the Rényi divergence between $M(x)$ and $M(x')$:

**Definition 2.2** (Rényi Divergence (Rényi, 1961)). Let $P$ and $Q$ be probability distributions. For $\lambda \in (1, \infty)$ the Rényi divergence of order $\lambda$ between $P$ and $Q$ is defined as

$$D_\lambda(P\|Q) = \frac{1}{\lambda - 1} \log\left(\mathop{\mathbb{E}}_{y \sim P}\left[\frac{P(y)^{\lambda-1}}{Q(y)^{\lambda-1}}\right]\right).$$

and for $\lambda = 1$, we define $D_1(P\|Q) = D_{\mathrm{KL}}(P\|Q)$,

**Definition 2.3** (Rényi differential privacy (Mironov, 2017)). A randomized mechanism $M : \mathcal{X}^n \to \mathcal{Y}$ is $(\lambda, \varepsilon)$-Rényi differentially private ($(\lambda, \varepsilon)$-RDP) if for any two neighboring datasets $x$ and $x'$,

$$D_\lambda(M(x)\|M(x')) \leq \varepsilon.$$

Intuitively, $\varepsilon$ controls the moments of the privacy loss random variable: $Z := \log \frac{P[M(x)=Y]}{P[M(x')=Y]}$, where $Y$ is distributed as $M(x)$, up to order $\lambda$. A smaller $\varepsilon$ and larger $\lambda$ correspond to a stronger privacy guarantee.

The following composition property of RDP mechanisms allow us to track the privacy guarantees of using multiple Dirichlet mechanisms. This can be helpful when Dirichlet mechanisms is employed in a more complex algorithms, such as fitting a discrete probabilistic model.

**Lemma 1** (Composition of RDP mechanisms (Mironov, 2017)). *Let $M_1 : \mathcal{X}^n \to \mathcal{Y}$ be a $(\lambda_1, \varepsilon_1)$-RDP mechanism and $M_2 : \mathcal{X}^n \to \mathcal{Z}$ be a $(\lambda_2, \varepsilon_2)$-RDP mechanism. Then a mechanism $M : \mathcal{X}^n \to \mathcal{Y} \times \mathcal{Z}$ defined by $M(x) = (M_1(x), M_2(x))$ is $(\min(\lambda_1, \lambda_2), \varepsilon_1 + \varepsilon_2)$-RDP.*

### 2.2 Exponential mechanism with the KL divergence

The exponential mechanism (McSherry & Talwar, 2007) is a privacy mechanism that releases an element from a range $\mathcal{Y}$ that approximately minimizes a given *loss function* $\ell : \mathcal{X}^N \times \mathcal{Y} \to \mathbb{R}$. Given a base measure $\mu$ over $\mathcal{Y}$ and a dataset $x \in \mathcal{X}^N$, the mechanism outputs $y \in \mathcal{Y}$ with probability density proportional to:

$$e^{-\beta\ell(x,y)}\mu(y), \tag{2}$$

where $\beta$ is a function of $\varepsilon$, the privacy parameter.

For the first time, we point out the connection between the exponential mechanism and a well-known family of probability distributions under a specific choice of $\ell(x, y)$. Let $f : \mathcal{X}^N \to \mathbb{R}_{\geq 0}^d$ be an arbitrary vector-valued

function on datasets. Let $\mathcal{Y} = S^{d-1}$. Assuming that $N_f := \sum_i f_i(x)$ is known and nonzero, we denote the normalized vector $\widetilde{f(x)} = N_f^{-1} f(x) \in S^{d-1}$. In equation 2, let $\ell(x, y) = D_{\mathrm{KL}}(\widetilde{f(x)} \| y)$, $\beta = r N_f$, and $\mu(y)$ be the density of Dirichlet($\boldsymbol{\alpha}$), that is, $\mu(y) \propto \prod_{i=1}^d y_i^{\alpha-1}$. Then, the probability density of the output $y$ of the corresponding exponential mechanism is proportional to:

$$\exp\left(-r N_f D_{\mathrm{KL}}(\widetilde{f(x)} \| y)\right) \prod_i y_i^{\alpha-1} = \exp\left(r \sum_{i, x_i \neq 0} f_i(x) \log(y_i / \widetilde{f_i(x)})\right) \prod_i y_i^{\alpha-1}$$

$$\propto \prod_{i, x_i \neq 0} y_i^{r f_i(x)} \prod_i y_i^{\alpha-1}$$

$$= \prod_i y_i^{r f_i(x) + \alpha - 1},$$

which is exactly the non-normalized density function of Dirichlet($r f(x) + \alpha$). This specific distribution will play a major role in the main privacy mechanism introduced in the next section.

From this derivation, we can see that this particular instance of the exponential mechanism can be used to output $y$ that approximately minimizes the KL divergence $D_{\mathrm{KL}}\left(\widetilde{f(x)} \| y\right)$ while keeping $x$ private.

To see how the choices of $r$ and $\alpha$ affect the "distance" between $y_i$ and $\widetilde{f_i(x)}$, we treat $y_i$ as an estimator of $\widetilde{f_i(x)}$ and look at the bias of $y_i$:

$$\left| \mathbb{E}[y_i] - \widetilde{f_i(x)} \right| = \left| \frac{r f_i(x) + \alpha}{r N_f + d\alpha} - \frac{f_i(x)}{N_f} \right| = \frac{\alpha |N_f - d f_i(x)|}{N_f (r N_f + d\alpha)}. \tag{3}$$

The bias is reduced when $r$ increases and $\alpha$ decreases. We can also look at the variance of $y_i$:

$$\mathrm{Var}[y_i] = \frac{(r f_i(x) + \alpha)(r(N_f - f_i(x)) + (d-1)\alpha)}{(r N_f + d\alpha)^2 (r N_f + d\alpha + 1)},$$

which is $O(1/r)$ as $r \to \infty$ and $O(1/\alpha)$ as $\alpha \to \infty$. This implies that draws from Dirichlet($r f(x) + \alpha$) are more concentrated when $r$ and $\alpha$ are large.

**Applications.**   The derivation of the Dirichlet mechanism above suggests that the best use of the Dirichlet mechanism is for privately minimizing KL divergence, which arises in the following scenarios:

1. **Maximum likelihood estimation.** Consider a problem of parameter estimation in a multinomial model with $d$ possible outcomes. Let $x \in [d]^N$ be $N$ observations, $f_1(x), \ldots, f_d(x)$ be the frequencies and $y_1, \ldots, y_d$ be the model's parameters. Then the log-likelihood of $x$ is $\sum_i f_i(x) \log y_i$. Maximizing the log-likelihood with respect to $y$ is equivalent to minimizing the KL divergence:

$$\arg\max_y \sum_i f_i(x) \log y_i = \arg\min_y D_{\mathrm{KL}}\left(\frac{f(x)}{N} \middle\| y\right).$$

   Thus, we can use the Dirichlet mechanism to release the parameters of the model while keeping $x$ private.

2. **Cross-entropy minimization.** Consider the same multinomial model as above. One might instead aim to minimize the cross-entropy loss: $-\frac{1}{N} \sum_i f_i(x) \log y_i$ over $y$. This is also equivalent to minimizing the KL divergence, so we can use the Dirichlet mechanism to privately solve for $y$.

3. **Private estimation of a discrete distribution.** If we further assume that $x$ is a sample from an *unknown* discrete distribution $p \in S^{d-1}$ with $p_i > 0$ for all $i$, a single draw $y \sim \text{Dirichlet}(r f(x) + \alpha)$ can be used to privately estimate $p$ in KL divergence. The KL divergence between $p$ and $y$ can be bounded as follows:

$$D_{\mathrm{KL}}(p\|y) = D_{\mathrm{KL}}(\widetilde{f(x)}\|y) - D_{\mathrm{KL}}(\widetilde{f(x)}\|p) + \sum_i (p_i - \widetilde{f_i(x)}) \log(p_i/y_i)$$

$$\leq D_{\mathrm{KL}}(\widetilde{f(x)}\|y) + \max_i |\log(p_i/y_i)| \sum_i |\widetilde{f_i(x)} - p_i|. \tag{4}$$

Here, we sketch a proof that, with high probability, the bound is small for a sufficiently large $N_f$. Due to Theorem 2 below and Agrawal (2020, Theorem I.2), respectively, both $D_{\mathrm{KL}}(\widetilde{f(x)}\|y)$ and $D_{\mathrm{KL}}(\widetilde{f(x)}\|p)$ are small w.h.p. Combining this with Pinsker's inequality: $\sum_i |\widetilde{f_i(x)} - q_i| \leq \sqrt{2 D_{\mathrm{KL}}(\widetilde{f(x)}\|q)}$, with $q = y$ and $q = p$, we obtain $y_i \approx \widetilde{f_i(x)} \approx p_i$, and so the second term is also small w.h.p. Note that we also have a similar bound for $D_{\mathrm{KL}}(y\|p)$ by switching $y$ and $p$ in equation 4. However, if some of the $p_i$'s are really small, it will take a large number of data points to bound the logarithmic term in equation 4. Finding finite sample bounds for $D_{\mathrm{KL}}(p\|y)$ and $D_{\mathrm{KL}}(y\|p)$ is an interesting problem that we leave open for further investigation.

### 2.3 Polygamma functions

In most of this study, we take advantage of several nice properties of the log-gamma function and its derivatives. The *polygamma function of order* $m$ is the $(m+1)$-th derivative of the logarithm of the gamma function. Specifically, when $m = 0$, we have the *digamma function* $\psi(x) := \frac{d}{dx} \log \Gamma(x)$, which is a concave and increasing function.

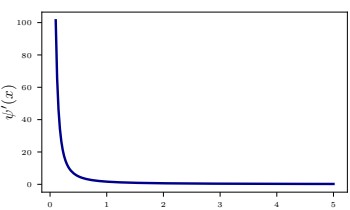

Figure 1: A plot of $\psi'(x)$.

Our function of interest is the polygamma function of order 1: $\psi'(x)$, which is a positive, convex, and decreasing function (see Figure 1). It has the series representation:

$$\psi'(x) = \sum_{k=0}^{\infty} \frac{1}{(x+k)^2}, \tag{5}$$

which allows for fast approximations of $\psi'(x)$ at any precision. $\psi'$ can also be approximated by the reciprocals:

$$\frac{1}{x} + \frac{1}{2x^2} < \psi'(x) < \frac{1}{x} + \frac{1}{x^2}, \tag{6}$$

which implies that $\psi'(x) \approx \frac{1}{x^2}$ as $x \to 0$ and $\psi'(x) \approx \frac{1}{x}$ as $x \to \infty$.

### 2.4 Related work

There are several studies on the differential privacy of obtaining a single draw from a probability distribution whose probability density function is of the form $y \mapsto \frac{1}{Z} p(x|y)\mu(y)$. Here, $x$ is sensitive data, $x \mapsto p(x|y)$ is a probability density function for all $y$ in the domain, $\mu$ is any positive-valued function, and $Z$ is the normalizing constant. Wang et al. (2015) showed that, when $|\log p(x \mid y)| \leq B$ for some constant $B$, then a single draw is $4B$-differentially private. However, the densities that we study are not bounded away from zero; they have the form $\prod_i y_i^{r f_i(x) + \alpha}$ which becomes small when one of the $y_i$'s is close to zero. Dimitrakakis et al. (2017) showed that, when $p$ is the density of the binomial distribution and $\mu$ is the density of the beta distribution, then a single draw is $(0, \delta)$-DP, and the result cannot be improved unless the parameters are assumed to be above a positive threshold. As a continuation of their work, we prove in the appendix that, when the parameters are bounded below by $\alpha > 0$, sampling from the Dirichlet distribution (which is a generalization of the beta distribution) is $(\varepsilon, \delta)$-DP with $\varepsilon > 0$.

Let $x$ be a sufficient statistic of an exponential family with finite $\ell^1$-sensitivity. Foulds et al. (2016) showed that sampling $Y \sim p(y \mid \hat{x})$, where $\hat{x} = x + $ Laplace noise, is differentially private and as asymptotically efficient as sampling from $p(y \mid x)$. However, for a small sample size, the posterior over the noisy statistics might be too far away from the actual posterior. Bernstein & Sheldon (2018) thus proposed to approximate the joint distribution $p(y, x, \hat{x})$ using Gibbs sampling, which is then integrated over $x$ to obtain a more accurate posterior over $\hat{x}$.

Geumlek et al. (2017) were the first to study sampling from exponential families with Rényi differential privacy (RDP; Mironov (2017)). Even though they provided a general framework to find $(\lambda, \varepsilon)$-RDP guarantees for exponential families, explicit forms of $\lambda$ and the upper bound of $\lambda$ were not given.

The privacy of data synthesis via sampling from Multinomial($Y$), where $Y$ is a discrete distribution drawn from the Dirichlet posterior, was first studied by Machanavajjhala et al. (2008). They showed that the data synthesis is $(\varepsilon, \delta)$-DP, where $\varepsilon$ grows by the number of draws from Multinomial($Y$). In contrast, we show that a single draw from the Dirichlet posterior is approximate DP, which by the post-processing property allows us to sample from Multinomial($Y$) as many times as we want while retaining the same privacy guarantee.

Gohari et al. (2021) have recently showed that the Dirichlet mechanism is $(\hat\varepsilon(r, \gamma, \eta, \eta'), \delta(r, \gamma, \eta, \eta'))$-DP, where $\gamma, \eta, \eta' \in (0,1)$ are additional parameters. Not only the guarantee has many parameters to optimize, it is also computational intensive. Specifically, for any $W \subset [d]$, define $\Omega_W^{\eta,\eta'} = \left\{ p \in S^{d-1} : p_i > \gamma, \forall i \in W; \sum_{i \in W} p_i \leq 1 - \eta' \right\}$. Gohari et al. proposed $\hat\varepsilon = \Theta(r \log(1/\gamma))$ and

$$\delta = 1 - \min_{p \in \Omega_W^{\eta,\eta'}, W \subset [d]} \left\{ \Pr[Y_i > \gamma; \forall i \in W] : Y \sim \text{Dirichlet}(rp) \right\}. \tag{7}$$

To compute $\delta$, we have to approximate $\Pr[Y > \gamma]$ with a numerical integration scheme with high precision, otherwise the integral may be greater than one. Even then, the integral is highly dependent on the scheme, and for some choices of the parameters $r, \eta, \eta'$, the value of $\delta$ cannot go below a certain threshold. We illustrate this in Figure 2. With $r = 171.87, \eta = 0.028$ and $\eta' = 0.114$, the value of $\delta$ cannot go below $2.1 \times 10^{-4}$. In contrast, our guarantee is much simpler to compute, as the function $\psi'$ can be easily approximated via its series representation (equation 5). Moreover, we are the first to provide the utility of the Dirichlet mechanism in terms of KL divergence minimization.

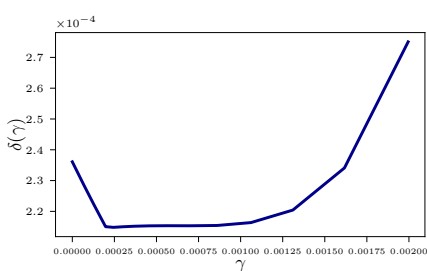

Figure 2: A numerical simulation of $\delta$ (equation 7) as a function of $\gamma$.

## 3 Main privacy mechanism

### 3.1 The Dirichlet mechanism

Let $f : \mathcal{X}^N \to \mathbb{R}_{\geq 0}^d$ be an arbitrary vector-valued function with finite $\ell^2$- and $\ell^\infty$-sensitivities: there exist two constants $\Delta_2, \Delta_\infty > 0$ such that

$$\sup_{x, x' \text{ neighboring}} \|f(x) - f(x')\|_2^2 \leq \Delta_2^2 \quad \text{and} \quad \sup_{x, x' \text{ neighboring}} \|f(x) - f(x')\|_\infty \leq \Delta_\infty.$$

Algorithm 1 below details the Dirichlet mechanism used to privatize $x \in \mathcal{X}^N$.

---
**Algorithm 1** $(\lambda, \varepsilon)$-RDP Dirichlet mechanism
---
**Input:** A dataset $x \in \mathcal{X}^N$, A vector-valued function $f : \mathcal{X}^N \to \mathbb{R}_{\geq 0}^d$ with $\ell^2$-sensitivity $\Delta_2$ and $\ell^\infty$-sensitivity $\Delta_\infty$
**Parameters:** $\lambda \geq 1$, $\varepsilon > 0$
1. Use a root-finding algorithm to find $r > 0$ such that $\varepsilon = \frac{1}{2}\lambda r^2 \Delta_2^2 \psi'(1 + 3(\lambda - 1)r\Delta_\infty)$.

2. Let $\alpha = 1 + 4(\lambda - 1)r\Delta_\infty$.

3. Output $y \sim \text{Dirichlet}(rf(x) + \alpha)$.

---

The following lemma ensures that we can obtain an $r > 0$ in Line 1 for any $\varepsilon > 0$:

**Lemma 2.** *With $\varepsilon, \Delta_2 > 0, \Delta_\infty > 0$ and $\lambda \geq 1$ held constant, the function $r \mapsto \frac{1}{2}\lambda r^2 \Delta_2^2 \psi'(1 + 3(\lambda - 1)r\Delta_\infty)$ defined on $(0, \infty)$ is strictly increasing from 0 to $\infty$. Consequently, the equation*

$$\varepsilon = \frac{1}{2}\lambda r^2 \Delta_2^2 \psi'(1 + 3(\lambda - 1)r\Delta_\infty)$$

*has a unique solution in $r$ for any $\varepsilon, \Delta_2, \Delta_\infty > 0$ and $\lambda \geq 1$.*

The proof of Lemma 2 can be found in Appendix D.

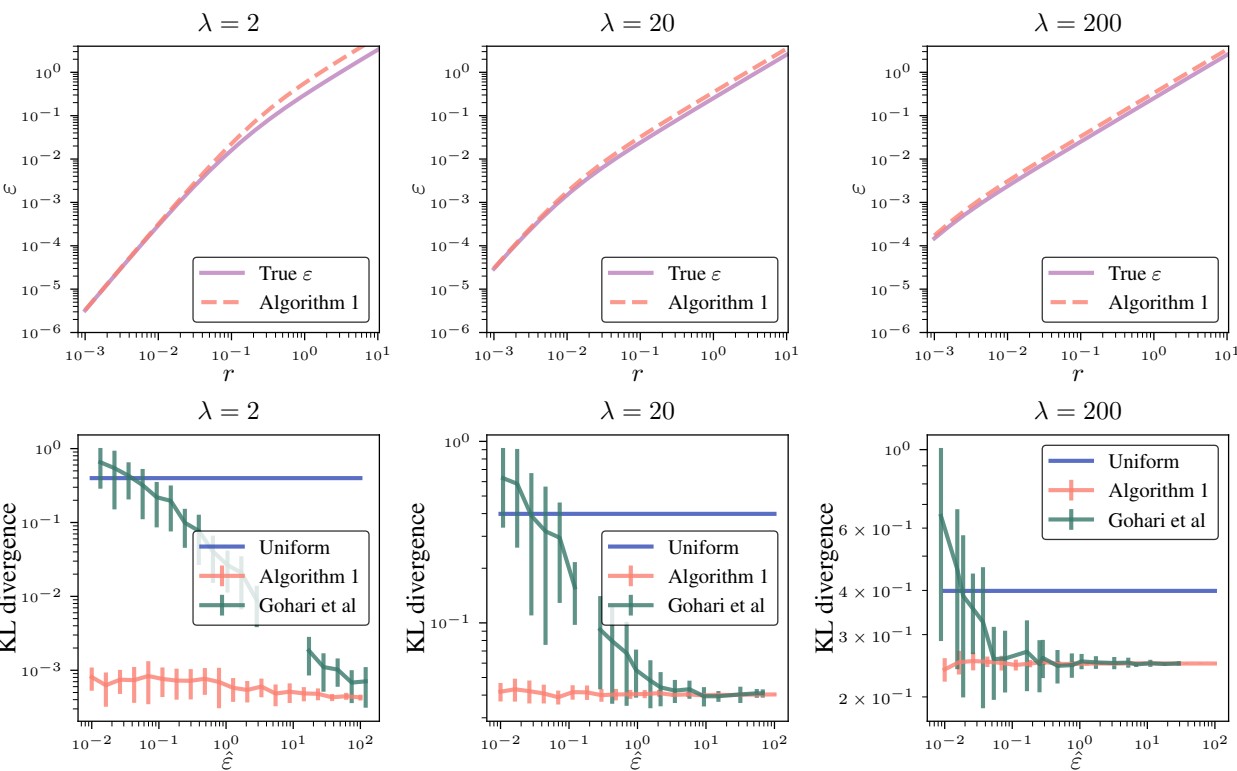

Figure 3: Top: Plots of the Rényi divergence ($\varepsilon$) between Dirichlet($rf(x) + \alpha$) and Dirichlet($rf(x') + \alpha$) using the direct calculations and Algorithm 1 as a function of $r$ for $\lambda \in \{2, 20, 200\}$. Here, $f(x) = (11, 8, 65, 25, 38, 1)$, $f(x') = (11, 7, 65, 25, 38, 0)$ and $\alpha = 1 + 4(\lambda - 1)r$. Bottom: Plots of $D_{\mathrm{KL}}(y \| \widetilde{f(x)})$ for multiple instances of $y$ drawn from Dirichlet($rf(x) + \alpha$), where $f(x) = (119, 74, 618, 272, 13, 187)$, $\alpha = 1 + 4(\lambda - 1)r$, and $\lambda \in \{2, 20, 200\}$. For each $\hat{\varepsilon}$, the privacy parameter $r$ is chosen to satisfy ($\hat{\varepsilon}, 10^{-5}$)-DP according to (1) the results of Gohari et al. (2021), and (2) the conversion from our RDP guarantee to approximate DP.

## 3.2 Privacy guarantee

**Theorem 1.** *Algorithm 1 is $(\lambda, \varepsilon)$-RDP.*

The proof of Theorem 1 can be found in Appendix E. A few remarks are in order.

**Remark 1.** In general, we can replace $\psi'(1 + 3(\lambda - 1)r\Delta_\infty)$ in Line 1 by $\psi'(1 + g(r))$, and $\alpha = 1 + 4(\lambda - 1)r\Delta_\infty$ in Line 2 by $\alpha = 1 + g(r) + (\lambda - 1)r\Delta_\infty$ for any function $g : \mathbb{R}_{>0} \to \mathbb{R}_{\geq 0}$. In particular, choosing $g \equiv 0$ yields $r = \sqrt{2\varepsilon/(\lambda\Delta_2^2\psi'(1))}$ which can be computed without a root-finding algorithm. However, this choice of $r$ makes $\varepsilon$ grows as $r^2$, which becomes too large when $r > 1$. Instead, we choose $g(r)$ to be a constant factor of an existing term $(\lambda - 1)r\Delta_\infty$ in $\alpha$, which allows us to offset the $\lambda r^2$ factor in $\varepsilon$ with $\psi'(1 + g(r)) = \Theta\left(\frac{1}{1+(\lambda-1)r}\right)$.

**Remark 2.** If one has prior knowledge that $f_i(x) > b$ for some $b > 0$ for all $x \in \mathcal{X}^N$ and all $i \in [d]$, then the proof of Theorem 1 can be modified so that $(\lambda, \varepsilon)$-RDP can be obtained by setting $r$ to be the solution to the equation $\varepsilon = \frac{1}{2}\lambda r^2 \Delta_2^2 \psi'(1 + rb + 3(\lambda - 1)r\Delta_\infty)$. Since $\psi'$ is strictly decreasing, this leads to a larger value of $r$ compared to Algorithm 1.

To demonstrate the tightness of the privacy guarantee of Algorithm 1, we simulate two neighboring histograms: $f(x) = (11, 8, 65, 25, 38, 1)$ and $f(x') = (11, 7, 65, 25, 38, 0)$. As functions of $r$, we compare $\varepsilon$ in Line 1 with the analytic values of the Rényi divergence between Dirichlet($rf(x) + \alpha$) and Dirichlet($rf(x') + \alpha$), where $\alpha$ is given in Line 2. The plots of $\varepsilon$ as functions of $r$ in Figure 3 show that our proposed RDP-guarantees are close to the actual Rényi divergences across different values of $\lambda$.

We also perform another simulation in order to compare our privacy guarantees with the ones from Gohari et al. (2021) in terms of their effects on the KL divergence. In this simulation, we apply the Dirichlet mechanism with these privacy guarantees to the following count data: $f(x) = (119, 74, 618, 272, 13, 187)$. For each $\lambda \in \{2, 20, 200\}$, we define $\alpha = 1 + 4(\lambda - 1)r$ as in Algorithm 1. Since the results of Gohari et al. are stated in terms of approximate DP, we have to convert our result from RDP to approximate DP (see Appendix B for more details on the conversion). For each $\hat{\varepsilon}$ ranging from 0.001 to 100, we use Theorem 1 (with the conversion) and Gohari et al.'s results to choose $r > 0$ so that a single draw from Dirichlet$(rf(x) + \alpha)$ is $(\hat{\varepsilon}, 10^{-5})$-DP. We then draw multiple instances, say $y$, from the distribution and compute $D_{\mathrm{KL}}(\widetilde{f(x)} \| y)$. Finally, we plot the KL divergence as a function of $\hat{\varepsilon}$, as shown in Figure 3. As a baseline, we also plot the KL divergence between $\widetilde{f(x)}$ and the discrete uniform distribution. We can see that our privacy guarantee generally provides smaller KL divergences than that of Gohari et al.'s. However, as $\lambda$ becomes very large, the algorithms output discrete probability distributions that are close to being uniform. The missing points in the $\lambda = 2$ and $\lambda = 20$ plots are related to a precision issue with the Gohari et al.'s method that we pointed out in Section 2.4: because of insufficient precision in numerical integration, we could not bring the value of $\delta$ down to $10^{-5}$.

## 4 Utility

Let us recap the setting with which we apply the Dirichlet mechanism: we have a sensitive dataset $x \in \mathcal{X}^N$ and an arbitrary vector-valued function $f : \mathcal{X}^N \to \mathbb{R}^d_{\geq 0}$. Let $N_f := \sum_i f_i(x)$ and $\widetilde{f(x)} := N_f^{-1} f(x) \in S^{d-1}$. We propose the Dirichlet mechanism (Algorithm 1) which aims to output $y$ that minimizes $D_{\mathrm{KL}}\left(\widetilde{f(x)} \| y\right)$ while keeping $x$ private. This motivates us to measure the utility of the Dirichlet mechanism in terms of the KL divergence between $\widetilde{f(x)}$ and $y$. To this end, we can make use of the following bound:

**Theorem 2.** *For any $\alpha > 0$, $p = (p_1, \ldots, p_d) \in S^{d-1}$ and $q \sim \mathrm{Dirichlet}(\beta p + \alpha)$, the following inequality holds for any $\eta > 0$ and any $\beta \geq d\alpha/(e^{\eta/2} - 1)$:*

$$\Pr[D_{\mathrm{KL}}(p\|q) > \eta] \leq e^{-\beta \eta^2 / (2(2+\eta)(4+3\eta))}.$$

The proof can be found in Appendix F. Since the Dirichlet mechanism outputs $y \sim \mathrm{Dirichlet}(rf(x) + \alpha) = \mathrm{Dirichlet}(rN_f \widetilde{f(x)} + \alpha)$, we can apply Theorem 2 with $p = \widetilde{f(x)}$, $q = y$ and $\beta = rN_f$. As long as $N_f \geq d\alpha/\left(r(e^{\eta/2} - 1)\right)$, we have the bound

$$\Pr\left[D_{\mathrm{KL}}\left(\widetilde{f(x)} \| y\right) > \eta\right] \leq e^{-rN_f \eta^2 / (2(2+\eta)(4+3\eta))}.$$

We shall assume that $\eta \ll 1$ and $\lambda \geq 2$. To obtain $D_{\mathrm{KL}}\left(\widetilde{f(x)} \| y\right) > \eta$ with high probability, one needs $N_f = \Omega\left(\frac{1}{r\eta^2} + \frac{d\alpha}{r(e^{\eta/2}-1)}\right)$. Now, we would like to write $r$ and $\alpha$ in terms of $\varepsilon$ and $\lambda$ using the following identities from Algorithm 1.

$$\varepsilon = \frac{1}{2}\lambda r^2 \Delta_2^2 \psi'(1 + 3(\lambda - 1)r\Delta_\infty) \tag{8}$$

$$\alpha = 1 + 4(\lambda - 1)r\Delta_\infty. \tag{9}$$

We recall from Lemma 2 that the right-hand side of equation 8 is a strictly increasing function of $r$ from 0 to $\infty$. This implies that, as $\varepsilon \to \infty$, we have $r \to \infty$. Under this limit, it follows from equation 6 that $\psi'(1 + 3(\lambda - 1)r\Delta_\infty) = \Theta\left(\frac{1}{(\lambda-1)r}\right)$. Thus, equation 8 and 9 give $r = \Theta(\varepsilon)$ and $\alpha = \Theta((\lambda - 1)\varepsilon)$. On the other hand, as if $\varepsilon \to 0$, we have $r \to 0$ which implies $\psi'(1 + 3(\lambda - 1)r\Delta_\infty) = \Theta(1)$. Consequently, $r = \Theta(\sqrt{\varepsilon/\lambda})$ and $\alpha = \Theta(1)$. Therefore, to attain the $(\lambda, \varepsilon)$-RDP guarantee, one needs

$$N_f = \begin{cases} \Omega\left(\frac{1}{\varepsilon\eta^2} + \frac{d(\lambda-1)}{e^{\eta/2}-1}\right) & \text{if } \varepsilon \geq 1 \\ \Omega\left(\sqrt{\frac{\lambda}{\varepsilon}}\left[\frac{1}{\eta^2} + \frac{d}{e^{\eta/2}-1}\right]\right) & \text{if } \varepsilon < 1. \end{cases}$$

The most common example is when the data is categorical, that is, $x \in [d]^N$ and $f_i(x)$ is the number of $i$'s in $x$. Then $N_f = \sum_i f_i(x) = N$, and the analysis above implies that the sample complexity for $(\lambda, \varepsilon)$-RDP and sub-$\eta$ KL divergence, with $\lambda$ and $\eta$ fixed, is $N = \Omega\left(\frac{1}{\varepsilon} + 1\right)$ if $\varepsilon \geq 1$ and $N = \Omega\left(\frac{1}{\sqrt{\varepsilon}}\right)$ if $\varepsilon < 1$.

## 5 Experiments and discussions

### 5.1 Naïve Bayes classification

We consider the Dirichlet mechanism for differentially private multinomial naïve Bayes classification. Given a dataset $D = \{(x^{(i)}, y^{(i)})\}_{i=1}^N$, we construct a model to classify labels $y^{(i)} \in [d]$ from discrete features $x^{(i)} = (x_1^{(i)}, \ldots, x_K^{(i)}) \in \prod_{k=1}^K \mathcal{X}_k$, where $\mathcal{X}_1, \ldots, \mathcal{X}_K$ are finite sets. For $j \in [d]$, $k \in [K]$ and $c \in \mathcal{X}_k$, we denote the class count by $N_j := \sum_{i=1}^N \mathbb{I}(y^{(i)} = j)$. For the $k$-th feature, we denote the feature-class count by $N_{jc}^k := \sum_{i=1}^N \mathbb{I}(y^{(i)} = j, x_k^{(i)} = c)$. We can use the count data to estimate the class probabilities and the class-conditional feature probabilities:

$$\Pr[y = j] := \hat{\pi}_j = N_j/N \quad \text{and} \quad \Pr[x_k = c | y = j] := \hat{\theta}_{jc}^k = N_{jc}^k/N_j. \tag{10}$$

The naïve Bayes model assumes that, conditioning on the label, the features are independent. As a result, the probability of $y = j$ conditioned on $(x_1, \ldots, x_K)$ can be computed as follows:

$$\Pr[y = j | x_1, \ldots, x_K] \propto \Pr[y = j] \prod_{k=1}^K \Pr[x_k = c | y = j]$$

$$= \frac{N_j}{N} \prod_{k=1}^K \frac{N_{jx_k}^k}{N_j}$$

$$= \hat{\pi}_j \prod_{k=1}^K \hat{\theta}_{jx_k}^k.$$

To modify the model with the Dirichlet mechanism, we sample $(\tilde{\pi}_1, \ldots, \tilde{\pi}_d) \sim \text{Dirichlet}(r(N_1, \ldots, N_d) + \alpha)$, where $r$ and $\alpha$ are chosen according to Algorithm 1 (with $\Delta_2^2 = 2$ and $\Delta_\infty = 1$) to attain $(\lambda, \varepsilon/K + 1)$-RDP. Similarly, for each $k \in K$ and $c \in \mathcal{X}_k$, we sample $(\tilde{\theta}_{1c}^k, \ldots, \tilde{\theta}_{dc}^k) \sim \text{Dirichlet}(r_c^k(N_{1c}^k, \ldots, N_{dc}^k) + \alpha_c^k)$, where $r_c^k$ and $\alpha_c^k$ are chosen to attain $(\lambda, \varepsilon/(K+1))$-RDP as well. We then release $\tilde{\pi}_j$ instead of $\hat{\pi}_j$ and $\tilde{\theta}_{jc}^k$ instead of $\hat{\theta}_{jc}^k$ for all $j, k$ and $c$, which leads to $(\lambda, \varepsilon)$-RDP by the basic composition (Lemma 1) and the parallel composition of RDP mechanisms

To benchmark the Dirichlet mechanism, we apply the Gaussian mechanism and the Laplace mechanism to the naïve Bayes model. Specifically, we replace $N_j$ and $N_{jc}^k$ in equation 10 by their noisy versions, namely $\tilde{N}_j := N_j + z_j$ and $\tilde{N}_{jc}^k := N_{jc}^k + z_{jc}^k$ where $z_j, z_{jc}^k \sim \mathcal{N}(0, \lambda(K+1)/\varepsilon)$ for the Gaussian mechanism and $z_j, z_{jc}^k \sim \text{Laplace}(0, b)$, where $b$ is calculated using Mironov (2017, Corollary 2) to attain $(\lambda, \varepsilon/K)$-RDP for the Laplace mechanism.

In this experiment, the naïve Bayes models with differentially private mechanisms are used to classify 8 UCI datasets (Dua & Graff, 2017) with diverse number of instances/attributes/classes. The details of the datasets are shown in Table 1. For each dataset, we use a 70-30 train-test split. Before fitting the models, numerical attributes are transformed into categorical ones using quantile binning, where the number of bins is fixed at 10.

For all privacy mechanisms, we fix $\lambda = 5$ and study their performances as $\varepsilon$ increases from $10^{-3}$ to $10$. We also add the random guessing model, which is a $(\lambda, 0)$-RDP model, as the baseline. The classification performances, measured in cross-entropy (CE) loss and accuracy on the test sets, are shown in Figure 4 and 5. We can see that, on all datasets, the test CE losses of the Dirichlet mechanism are substantially less than those of the Gaussian mechanism and Laplace mechanism; they are remarkably close to those of the non-private model on the CreditCard, GermanCredit, Bank and Adult datasets. This result should not be surprising, as the Dirichlet mechanism is the exponential mechanism that aims to minimize the KL divergence, and thus the cross-entropy between the normalized counts and the parameters.

Table 1: UCI datasets used in the experiment

| Dataset | #Instances | #Attributes | #Classes | %Positive | Source |
|---|---|---|---|---|---|
| CreditCard | 30000 | 23 | 2 | 22% | Yeh & hui Lien (2009) |
| Thyroid | 7200 | 21 | 3 | − | Quinlan et al. (1986) |
| Shopper | 12330 | 17 | 2 | 15% | Sakar et al. (2018) |
| Digit | 5620 | 64 | 10 | − | Garris et al. (1997) |
| GermanCredit | 1000 | 20 | 2 | 30% | Grömping (2019) |
| Bank | 41188 | 20 | 2 | 11% | Moro et al. (2014) |
| Spam | 4601 | 57 | 2 | 39% | Cranor & LaMacchia (1998) |
| Adult | 48842 | 13 | 2 | 24% | Kohavi (1996) |

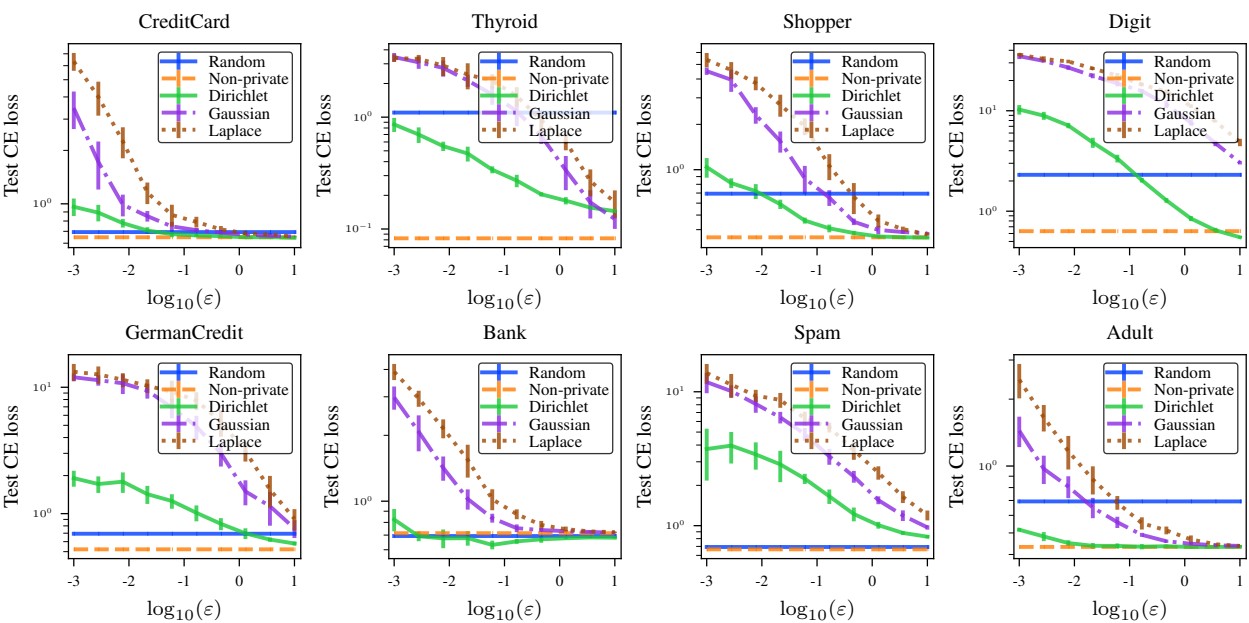

Figure 4: Test CE losses of the original and four $(5, \varepsilon)$-RDP naïve Bayes models on 8 UCI datasets.

In terms of accuracy, there is no clear winner among the three mechanisms; the Dirichlet mechanism performs as well as the other mechanisms in most cases. Specifically, it has higher accuracies than the Gaussian mechanism on the Digit dataset for $\varepsilon > 0.1$, on the Adult dataset for $\varepsilon < 0.1$, and on the Bank dataset for all values of $\varepsilon$.

The difference between the two metrics stem from the fact that the cross entropy loss is a continuous function of the predicted probability, while the accuracy is a result of applying a hard threshold on the probability. Thus the accuracy does not distinguish between, for example, two instances, $x, x'$ with $\Pr[y = 1|x] = 0.1$ and $\Pr[y = 1|x'] = 0.4$, but the CE loss will suffer almost three times as much when the true label of $x$ is 1 compared to when the true label of $x'$ is 1. Thus a model with high accuracy can have relatively low CE loss when they are too confident in their incorrect predictions.

All in all, neither metric is an end-all for measuring classification performance, and we should look at more than one metrics when fitting a model. If one wants to publish a naïve Bayes model under privacy constraint that performs well in both CE loss and accuracy, then the Dirichlet mechanism is an attractive option.

## 5.2 Parameter estimations of Bayesian networks

We use the Dirichlet mechanism for differentially private parameter estimations of discrete Bayesian networks. Consider a dataset $D = \{x^{(i)}\}_{i=1}^{N}$, where $x^{(i)} = (x_1^{(i)}, \ldots, x_K^{(i)}) \in \prod_{k=1}^{K} \mathcal{X}_k$ and $\mathcal{X}_1, \ldots, \mathcal{X}_K$ are finite sets. We

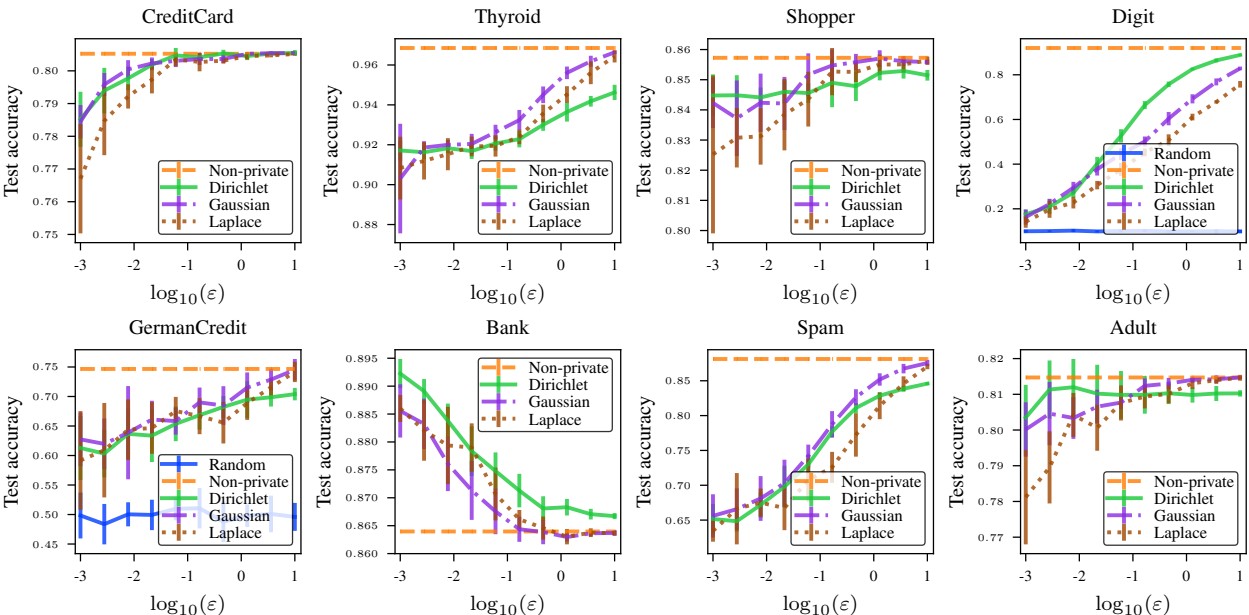

Figure 5: Test accuracies of the original and four $(5, \varepsilon)$-RDP naïve Bayes models on 8 UCI datasets. Plots of the random guessing on some datasets are not shown as its accuracies are well below the other models'.

name the $K$ variables by their index: $1, \ldots, K$. Given a Bayesian network and $k \in [K]$, we denote the set of parents of $k$, that is, the set of direct causes of $k$ by $Pa(k)$. Let $x_{Pa(k)}^{(i)} \coloneqq (x_\ell^{(i)})_{\ell \in Pa(k)}$ be observed values of $Pa(k)$ and $\mathcal{X}_{Pa(j)} \coloneqq \prod_{\ell \in Pa(j)} \mathcal{X}_\ell$ be the product space of $Pa(k)$. Given $j \in \mathcal{X}_k$ and $c \in \mathcal{X}_{Pa(k)}$, we denote $N_c^k \coloneqq \sum_{i=1}^N \mathbb{I}(x_{Pa(k)}^{(i)} = c)$ and $N_{jc}^k \coloneqq \sum_{i=1}^N \mathbb{I}(x_k^{(i)} = j, x_{Pa(k)}^{(i)} = c)$. The log-likelihood of the parameters $\theta_{jc}^k \coloneqq \Pr[x_k = j \mid x_{Pa(k)} = c]$ is given by:

$$LL(\theta) \coloneqq \sum_{k \in [K]} \sum_{\substack{j \in \mathcal{X}_k \\ c \in \mathcal{X}_{Pa(k)}}} N_{jc}^k \log \theta_{jc}^k. \tag{11}$$

Using the first-derivative test, the maximum-likelihood estimators of the Bayesian network are as follow:

$$\hat{\theta}_{jc}^k \coloneqq \frac{N_{jc}^k}{N_c^k}. \tag{12}$$

We can modify the model using the Dirichlet mechanism: assuming that $\mathcal{X}_k = [d]$, we replace $(\hat{\theta}_{1c}^k, \ldots, \hat{\theta}_{dc}^k)$ by $(\tilde{\theta}_{1c}^k, \ldots, \tilde{\theta}_{dc}^k) \sim \text{Dirichlet}(r(N_{1c}^k, \ldots, N_{dc}^k) + \alpha)$. Here, $r$ and $\alpha$ are chosen according to Algorithm 1 to attain $(\lambda, \varepsilon/K)$-RDP. By the basic composition (Lemma 1) and the parallel composition, releasing $\tilde{\theta}_{jc}^k$ for all $k \in [K]$, $j \in \mathcal{X}_k$ and $c \in \mathcal{X}_{Pa(k)}$ is $(\lambda, \varepsilon)$-RDP.

We will compare the Dirichlet mechanism with the Gaussian and Laplace mechanisms. In equation 12, we replace $N_{jc}^k$ by its noisy version: $\tilde{N}_{jc}^k \coloneqq N_{jc}^k + z_{jc}^k$, where $z_{jc}^k \sim \mathcal{N}(0, \lambda K/\varepsilon)$ for the Gaussian mechanism and $z_{jc}^k \sim \text{Laplace}(0, b)$, where $b$ is calculated using Mironov (2017, Corollary 2) to attain $(\lambda, \varepsilon/K)$-RDP for the Laplace mechanism. In addition, we replace $N_c^k$ by $\tilde{N}_c^k \coloneqq \sum_j \tilde{N}_{jc}^k$.

In this experiment, we have prepared Bayesian networks on the Adult, Bank and GermanCredit datasets, which are parts of full networks provided by Le Quy et al. (2022). The Bayesian networks are shown in Figure 6. As in the previous experiment, we use a 70-30 train-test split on each dataset, and continuous attributes are transformed into categorical attributes via quantile binning, with the number of bins fixed at 10.

For all privacy mechanisms, we fix $\lambda = 5$ and study their performances, in terms of the log-likelihoods of the privatized parameters on the test sets, as $\varepsilon$ increases from $10^{-3}$ to 10. The plot of the log-likelihoods

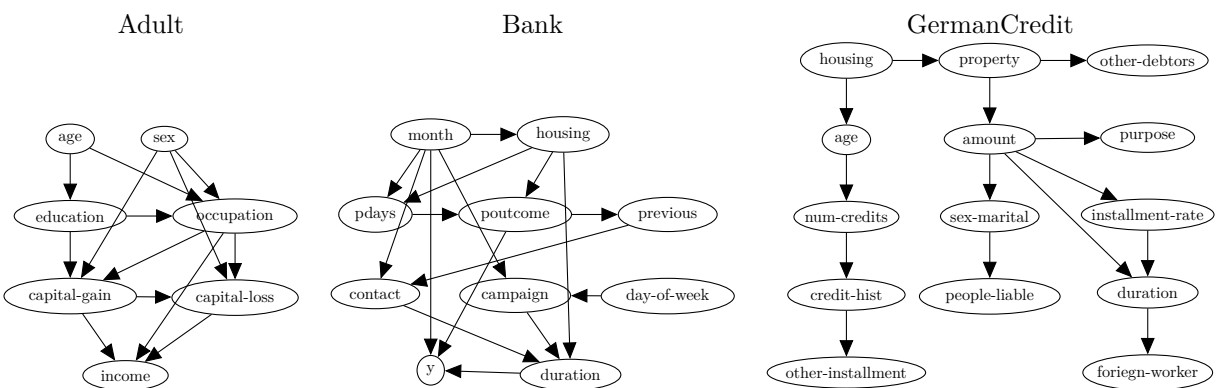

Figure 6: Our Bayesian networks on three datasets.

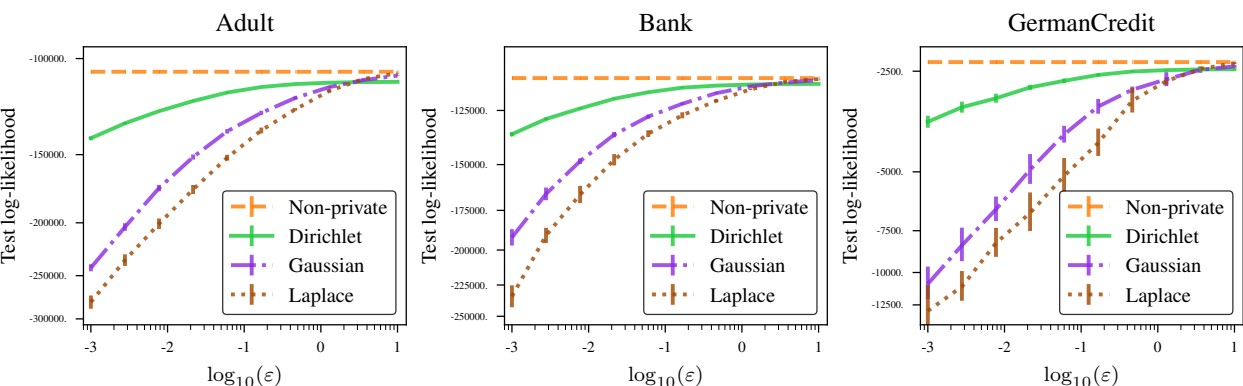

Figure 7: Test log-likelihoods of the parameters obtained from the maximum-likelihood estimation (non-private) and three $(5, \varepsilon)$-RDP mechanisms.

as functions of $\varepsilon$ are shown in Figure 7. We can see that, on all datasets, the test log-likelihoods of the Dirichlet mechanism are substantially less than those of the Gaussian mechanism and Laplace mechanism for $\varepsilon < 1$. The results agree with our suggestion to use the Dirichlet mechanism for privacy-aware KL divergence minimization for discrete parameters, as it is equivalent to likelihood maximization.

## 6 Conclusion

The Dirichlet mechanism is an instance of the exponential mechanism whose loss function is the discrete KL divergence—this motivates us to use the Dirichlet mechanism for private estimation of an empirical distribution in KL divergence. As a consequence, the Dirichlet mechanism can be used for private likelihood maximization and cross–entropy minimization. This work provides a choice for the multiplicative factor $r$ and the prior $\alpha$ that achieves a desired $(\lambda, \varepsilon)$-RDP guarantee. To demonstrate its efficiency, we compare our mechanism with the Gaussian and Laplace mechanisms for differentially private naïve Bayes classification, and as expected, the Dirichlet mechanism provides significantly lower cross-entropy losses on various datasets compared to the other two mechanisms. We also make a comparison between the mechanisms for maximum likelihood estimations for Bayesian networks. Our experiment on three datasets shows that the Dirichlet mechanism provides significantly higher log-likelihoods than the Gaussian and Laplace mechanisms.

As the KL divergence is a fundamental measure in information theory, we envision that the Dirichlet mechanism would become essential for many privacy-focused information-theoretic models with discrete parameters.

**Broader Impact Statement**

The Dirichlet mechanism does not provide privacy protection for free, but with a cost of some accuracy loss: the higher the privacy guarantee, the lower the accuracy of the privatized model compared to the original model. Any losses incurred from the inaccuracy must be taken into consideration before deploying the privatized model.

**Acknowledgments**

The author would like to thank the reviewers and the action editors for valuable comments and suggestions.

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

## A  Dirichlet posterior sampling is not $\varepsilon$-differentially private

We show that the Dirichlet posterior sampling does not satisfy the original notion of differential privacy—the pure differential privacy.

**Proposition 3.** *For any $\varepsilon > 0$, the mechanism that outputs $y \sim \text{Dirichlet}(rf(x) + \alpha)$ is not $\varepsilon$-differentially private.*

*Proof.* Without loss of generality, let $x = (0, 0, \ldots, 0)$ and $x' = (1, 0, \ldots, 0)$. Let $\alpha > 0$ be any positive number. Let $y \sim \text{Dirichlet}(rf(x) + \alpha)$ and $y' \sim \text{Dirichlet}(rf(x') + \alpha)$. For any $y_0 = (y_1, y_2, \ldots, y_d)$ with $\sum_i y_i = 1$, we have

$$\frac{\Pr[y = y_0]}{\Pr[y' = y_0]} = \frac{B(rf(x') + \alpha)}{B(rf(x) + \alpha)} \cdot \frac{\prod_i y_i^{rf_i(x) + \alpha}}{\prod_i y_i^{rf_i(x') + \alpha}}$$
$$= \frac{B(rf(x') + \alpha)}{B(rf(x) + \alpha)} \cdot \frac{1}{y_1}.$$

For any $\varepsilon > 0$, we can choose a sufficiently small $y_1 > 0$ so that the right-hand side is larger than $e^\varepsilon$.  $\square$

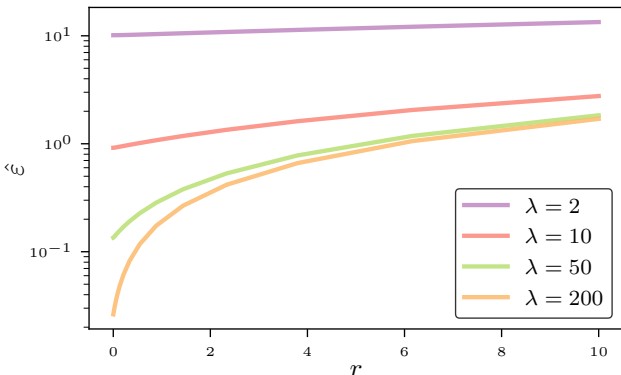

Figure 8: $(\varepsilon, \delta)$-DP guarantees of the Dirichlet mechanism following equation 14 with $\lambda \in \{2, 10, 50, 200\}$ and $\delta = 10^{-5}$.

## B  Approximate differential privacy

We can convert from RDP to approximate DP with the following conversion formula:

**Lemma 3** (From RDP to Approximate DP (Canonne et al., 2020)). *Let $\varepsilon > 0$. If $M$ is a $(\lambda, \varepsilon)$-RDP mechanism, then it also satisfies $(\hat{\varepsilon}, \delta)$-DP with*

$$\delta = \frac{\exp((\lambda - 1)(\varepsilon - \hat{\varepsilon}))}{\lambda - 1}\left(1 - \frac{1}{\lambda}\right)^{\lambda}. \tag{13}$$

Taking the logarithm of equation 13,

$$\log \delta = (\lambda - 1)(\varepsilon - \hat{\varepsilon}) + (\lambda - 1)\log(\lambda - 1) - \lambda \log(\lambda),$$

which is equivalent to

$$\hat{\varepsilon} = \varepsilon + \log(\lambda - 1) - \frac{\log \delta + \lambda \log(\lambda)}{\lambda - 1}.$$

Plugging in the RDP guarantee in Algorithm 1, we obtain

$$\hat{\varepsilon} = \frac{1}{2}\lambda r^2 \Delta_2^2 \psi'(1 + 3(\lambda - 1)r\Delta_\infty) + \log(\lambda - 1) - \frac{\log \delta + \lambda \log(\lambda)}{\lambda - 1}, \tag{14}$$

which gives a formula for $\hat{\varepsilon}$ in terms of $r$, $\lambda$ and $\delta$. Figure 8 shows $\hat{\varepsilon}$ as a function of $r$ at four different values of $\lambda$. We can see that, at a fixed $\delta$, $\hat{\varepsilon}$ is increased when we increase $r$ and decrease $\lambda$.

## C  Experiments with approximate DP

We perform the same experiments as those in Section 5. But this time, we focus on approximate DP instead of RDP, and we also include the Dirichlet mechanism with Gohari et al. (2021)'s privacy guarantee in the experiments. Our $(\lambda, \varepsilon)$-RDP guarantee of the Dirichlet mechanism is converted to $(\hat{\varepsilon}, \delta)$-DP guarantee, with $\delta = 10^{-5}$, using the material in Section sec:adp. The results of the naïve Bayes and Bayesian network experiments are shown in Figure 9 and Figure 10, and those of the Bayesian networks are shown in Figure 11.

Aside from similar results as those in Section 5, We highlight that our Dirichlet mechanism performs better than Gohari et al.'s in all experiments, and Gohari et al.'s mechanism performs significantly worse for smaller values of $\hat{\varepsilon}$. We also notice that, in contrast to the results in Section 5 the Laplace mechanism performs better than the Gaussian mechanism; this is because the composition property for multiple uses of an $\hat{\varepsilon}$ DP mechanism is better than that of an $(\hat{\varepsilon}, \delta)$-DP for any $\delta > 0$ (see Dwork & Roth (2014, Theorem 3.20)).

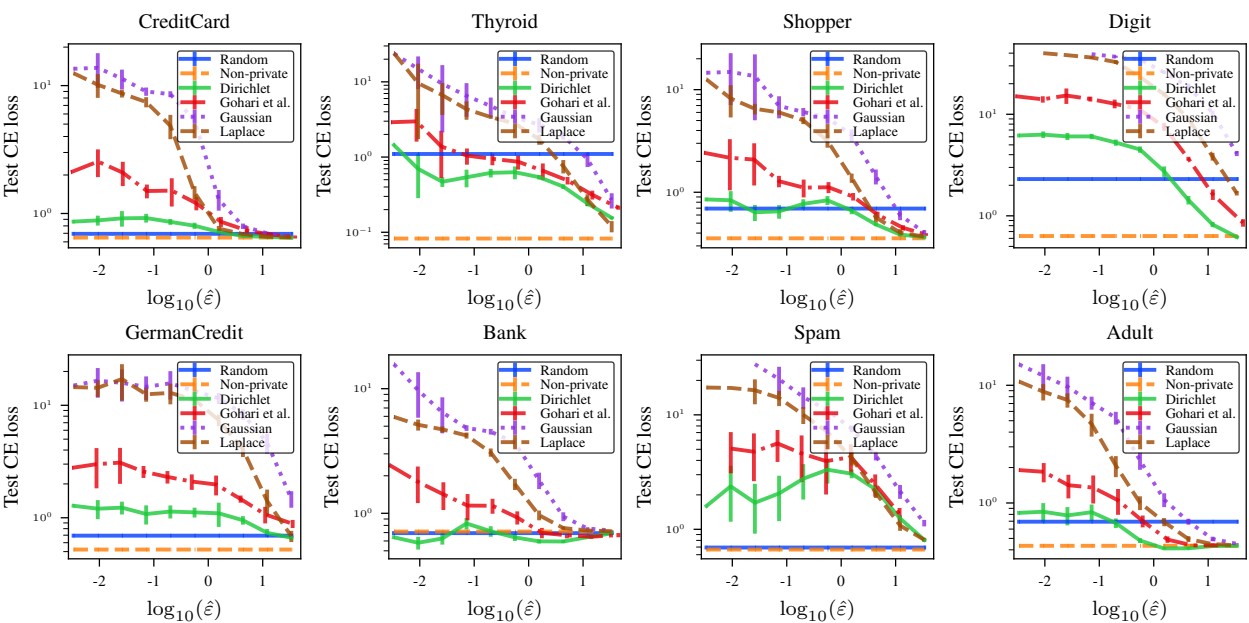

Figure 9: Test CE losses of the original and five $(\hat{\varepsilon}, 10^{-5})$-RDP naïve Bayes models on 8 UCI datasets.

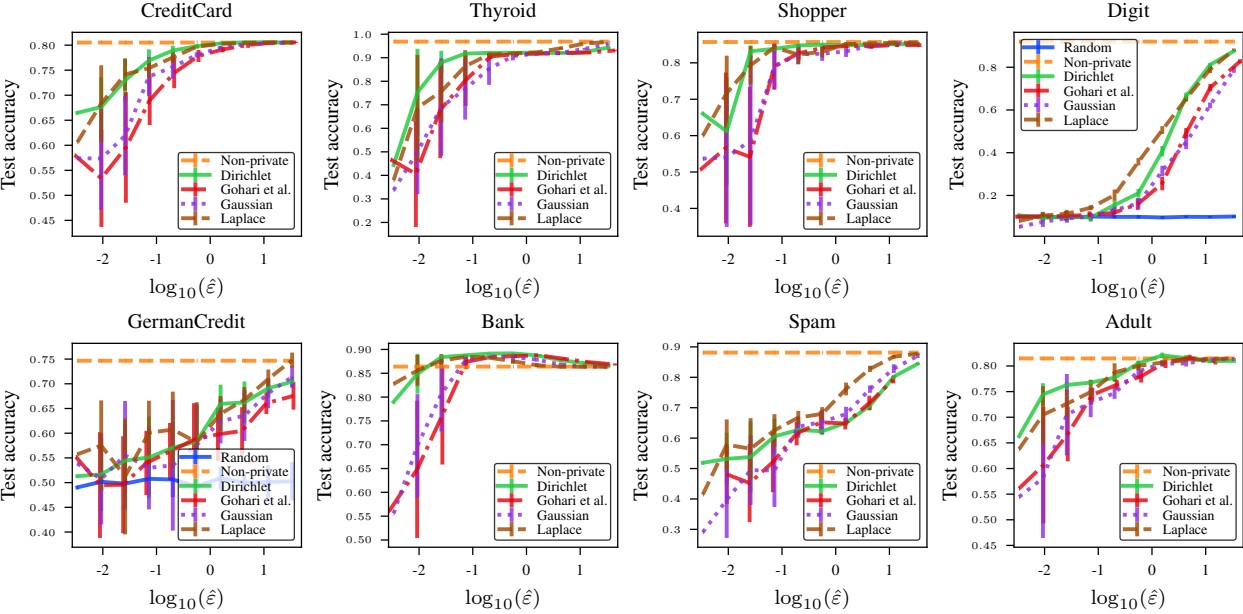

Figure 10: Test accuracies of the original and five $(\hat{\varepsilon}, 10^{-5})$-DP naïve Bayes models on 8 UCI datasets. Plots of the random guessing on some datasets are not shown as its accuracies are well below the other models'.

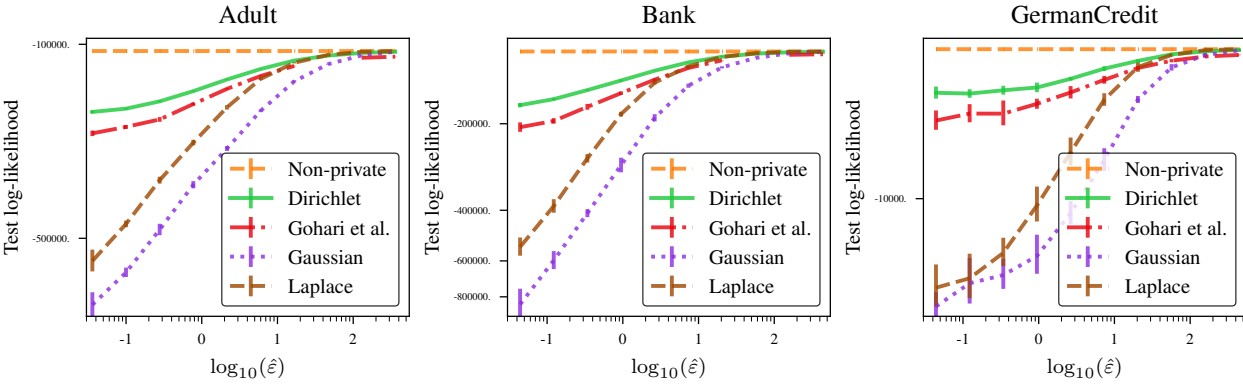

Figure 11: Test log-likelihoods of the parameters obtained from the maximum-likelihood estimation (non-private) and four $(\hat{\varepsilon}, 10^{-5})$-DP mechanisms.

## D Proof of Lemma 2

Denote $x = 3(\lambda - 1)r\Delta_\infty$. With $\varepsilon, \lambda, \Delta_2$ and $\Delta_\infty$ fixed as constants, we can write the equation as $\varepsilon = Cx^2\psi'(1 + x)$ for some constant $C > 0$. From equation 6, we have $\psi'(1 + x) = \Theta\left(\frac{1}{(1+x)^2}\right)$ as $x \to 0$ and $\psi'(x) = \Theta\left(\frac{1}{1+x}\right)$ as $x \to \infty$. Consequently,

$$\lim_{x \to 0} x^2\psi'(1 + x) = 0 \quad \text{and} \quad \lim_{x \to \infty} x^2\psi'(1 + x) = \infty. \tag{15}$$

The conclusion will follow if we can show that the function $\phi(x) := x^2\psi'(1 + x)$ is strictly increasing. For this, first we use $\psi'(1 + x) < \frac{1}{1+x} + \frac{1}{(1+x)^2}$ to obtain

$$[\psi'(1 + x)]^2 < \frac{\psi'(1 + x)}{1 + x} + \frac{\psi'(1 + x)}{(1 + x)^2} \leq \frac{2\psi'(1 + x)}{1 + x} < \frac{2\psi'(1 + x)}{x}.$$

In other words, $2\psi'(1 + x) > x[\psi'(1 + x)]^2$. Combining this with $[\psi'(x)]^2 + \psi''(x) > 0$ (see e.g. Batir (2004, Lemma1.1)), we have

$$\phi'(x) = 2x\psi'(1 + x) + x^2\psi''(1 + x) > x^2[\psi'(1 + x)]^2 + x^2\psi''(1 + x) = x^2\left([\psi'(x)]^2 + \psi''(x)\right) > 0.$$

Therefore, $\phi(x)$ is strictly increasing, which, combined with equation 15, implies that the equation $\phi(x) = \varepsilon$ has a unique solution $x_\varepsilon$ for any $\varepsilon > 0$. We then obtain a solution in $r$ by letting $r = x_\varepsilon/(3(\lambda - 1)\Delta_\infty)$.

## E Proof of Theorem 1

**Case 1:** $\lambda > 1$.

Let $x$ and $x'$ be neighboring datasets. For notational convenience, let $u := rf(x) + \alpha$ and $u' := rf(x') + \alpha$. As usual, we write $u = (u_1, \ldots, u_d)$, $u' = (u'_1, \ldots, u'_d)$, $u_0 := \sum_i u_i$ and $u'_0 := \sum_i u'_i$. Let $P(y)$ be the density of Dirichlet$(u)$ and $P'(y)$ be the density of Dirichlet$(u')$. To compute the Rényi divergence between $P(y)$ and $P'(y)$, we start with:

$$\mathbb{E}_{y \sim P(y)}\left[\frac{P(y)^{\lambda-1}}{P'(y)^{\lambda-1}}\right] = \frac{B(u')^{\lambda-1}}{B(u)^{\lambda-1}}\mathbb{E}_{y \sim P(y)}\left[y^{(\lambda-1)(u-u')}\right]$$

$$= \frac{B(u')^{\lambda-1}}{B(u)^{\lambda-1}} \cdot \frac{B(u + (\lambda - 1)(u - u'))}{B(u)}, \tag{16}$$

where $B(u) = \Gamma(u_0)^{-1}\prod_i \Gamma(u_i)$ is the multivariate beta function. Thus the ratio can be expressed in terms of gamma functions:

$$\frac{B(u')}{B(u)} = \frac{\prod_i \Gamma(u'_i)/\Gamma(\sum_i u'_i)}{\prod_i \Gamma(u_i)/\Gamma(\sum_i u_i)} = \frac{\Gamma(u_0)}{\Gamma(u'_0)}\prod_i \frac{\Gamma(u'_i)}{\Gamma(u_i)},$$

where $u_0 := \sum_i u_i$ and $u_0' := \sum_i u_i'$. Similarly,

$$\frac{B(u + (\lambda - 1)(u - u'))}{B(u)} = \frac{\Gamma(\sum_i u_i)}{\Gamma(\sum_i u_i + (\lambda - 1)\sum_i (u_i - u_i'))} \prod_i \frac{\Gamma(u_i + (\lambda - 1)(u_i - u_i'))}{\Gamma(u_i)}.$$

Taking the logarithm on both side of equation 16, we need to find an upper bound of:

$$\log \mathbb{E}_{y \sim P(y)}\left[\frac{P(y)^{\lambda - 1}}{P'(y)^{\lambda - 1}}\right] = \sum_i (G(u_i, u_i') + H(u_i, u_i')) - G(u_0, u_0') - H(u_0, u_0'), \tag{17}$$

where

$$G(u_i, u_i') := (\lambda - 1)(\log \Gamma(u_i') - \log \Gamma(u_i))$$
$$H(u_i, u_i') := \log \Gamma(u_i + (\lambda - 1)(u_i - u_i')) - \log \Gamma(u_i),$$

and similarly for $G(u_0, u_0')$ and $H(u_0, u_0')$. Using the second-order Taylor expansion, there exists $\xi$ between $u_i + (\lambda - 1)(u_i - u_i')$ and $u_i$, and $\xi'$ between $u_i$ and $u_i'$ such that

$$G(u_i, u_i') = -(\lambda - 1)(u_i - u_i')\psi(u_i) + \frac{1}{2}(\lambda - 1)(u_i - u_i')^2\psi'(\xi')$$

$$= -(\lambda - 1)(f_i(x) - f_i(x'))r\psi(u_i) + \frac{1}{2}(\lambda - 1)(f_i(x) - f_i(x'))^2 r^2 \psi'(\xi')$$

$$H(u_i, u_i') = (\lambda - 1)(u_i - u_i')\psi(u_i) + \frac{1}{2}(\lambda - 1)^2(u_i - u_i')^2\psi'(\xi)$$

$$= (\lambda - 1)(f_i(x) - f_i(x'))r\psi(u_i) + \frac{1}{2}(\lambda - 1)^2(f_i(x) - f_i(x'))^2 r^2 \psi'(\xi).$$

We try to find an upper bound of both $\psi'(\xi)$ and $\psi'(\xi')$. If $f_i(x) > f_i(x')$, then $u_i' < u_i < u_i + (\lambda - 1)(u_i - u_i')$. Thus both $\xi$ and $\xi'$ are bounded below by $u_i' \geq \alpha$. On the other hand, if $f_i(x) \leq f_i(x')$, then $u_i + (\lambda - 1)(u_i - u_i') \leq u_i \leq u_i'$. In this case, $\xi$ and $\xi'$ are bounded below by:

$$u_i + (\lambda - 1)(u_i - u_i') = f_i(x) + \alpha - (\lambda - 1)(rf_i(x') - rf_i(x))$$
$$\geq \alpha - (\lambda - 1)r\Delta_\infty.$$

Since $\psi'$ is decreasing, both $\psi'(\xi)$ and $\psi'(\xi')$ are bounded above by $\psi'(\alpha - (\lambda - 1)r\Delta_\infty)$. Consequently,

$$G(u_i, u_i') + H(u_i, u_i') \leq \frac{1}{2}\left((\lambda - 1) + (\lambda - 1)^2\right)(f_i(x) - f_i(x'))^2 r^2 \psi'(\alpha - (\lambda - 1)r\Delta_\infty)$$

$$= \frac{1}{2}\lambda(\lambda - 1)(f_i(x) - f_i(x'))^2 r^2 \psi'(\alpha - (\lambda - 1)r\Delta_\infty).$$

The same argument can be used to show that, there exist $\xi_0$ and $\xi_0'$ such that:

$$G(u_0, u_0') + H(u_0, u_0') = \frac{1}{2}(\lambda - 1)(u_0 - u_0')^2\psi'(\xi_0') + \frac{1}{2}(\lambda - 1)^2(u_0 - u_0')^2\psi'(\xi_0) > 0.$$

Therefore, continuing from equation 17,

$$D_\lambda(P(y)\|P'(y)) = \frac{1}{\lambda - 1}\left(\sum_i (G(u_i, u_i') + H(u_i, u_i')) - G(u_0, u_0') - H(u_0, u_0')\right)$$

$$< \frac{1}{\lambda - 1}\sum_i (G(u_i, u_i') + H(u_i, u_i'))$$

$$\leq \frac{1}{2}\lambda \sum_i (f_i(x_i) - f_i(x_i'))^2 r^2 \psi'(\alpha - (\lambda - 1)r\Delta_\infty)$$

$$\leq \frac{1}{2}\lambda \Delta_2^2 r^2 \psi'(\alpha - (\lambda - 1)r\Delta_\infty). \tag{18}$$

**Case 2:** $\lambda = 1$.

We use the following formula for the KL divergence between two Dirichlet distributions:

$$D_{\mathrm{KL}}(P(y)\|P'(y)) = \log\Gamma(u_0) - \sum_i \log\Gamma(u_i) - \log\Gamma(u_0')$$
$$+ \sum_i \log\Gamma(u_i') + \sum_i (u_i - u_i')(\psi(u_i) - \psi(u_0)),$$

From this, we split the right-hand side into two parts and apply the Taylor approximation as before:

$$-\sum_i \log\Gamma(u_i) + \sum_i \log\Gamma(u_i') + \sum_i (u_i - u_i')\psi(u_i) \leq \frac{1}{2}\sum_i (u_i - u_i')^2 \psi'(\min\{u_i, u_i'\})$$
$$\leq \frac{1}{2}\sum_i (u_i - u_i')^2 \psi'(1)$$
$$= \frac{1}{2}\sum_i (f_i(x_i) - f_i(x_i'))^2 r^2 \psi'(1)$$
$$\leq \frac{1}{2}\Delta_2^2 r^2 \psi'(1),$$

and

$$\log\Gamma(u_0) - \log\Gamma(u_0') - \sum_i (u_0 - u_0')\psi(u_0) \leq -\frac{1}{2}\sum_i (u_i - u_i')^2 \psi'(\max\{u_0, u_0'\})$$
$$\leq 0.$$

Adding these two inequalities yields the same inequality as equation 18 with $\lambda = 1$.

Thus, given any $\lambda \geq 1$, $\varepsilon > 0$ and any $g : \mathbb{R}_{>0} \to \mathbb{R}_{>0}$, if we let $r$ be the solution of $\frac{1}{2}\lambda r^2 \Delta_2^2 \psi'(1 + g(r)) = \varepsilon$ and $\alpha = 1 + g(r) + (\lambda - 1)r\Delta_\infty$, then the inequality above implies $D_\lambda(P(y)\|P'(y)) < \varepsilon$. We conclude that Algorithm 1 by setting $g(r) = 3(\lambda - 1)r\Delta_\infty$.

## F   Proof of the Utility bound

We first note a pair of inequalities for the digamma function, which hold for all $x > \frac{1}{2}$:

$$\log\left(x - \frac{1}{2}\right) < \psi(x) < \log x. \tag{19}$$

We start with the Chernoff bound: for any $t \leq \beta$,

$$\Pr[D_{\mathrm{KL}}(p\|q) > \eta] \leq e^{-t\eta}\mathbb{E}\left[e^{tD_{\mathrm{KL}}(p\|q)}\right]$$
$$= e^{-t\eta}\mathbb{E}\left[\prod_i (p_i/q_i)^{tp_i}\right]$$
$$= e^{-t\eta}\prod_i p_i^{tp_i}\mathbb{E}\left[\prod_i q_i^{-tp_i}\right]$$
$$= e^{-t\eta}\prod_i p_i^{tp_i}\frac{1}{B(\beta p + \alpha)}\int \prod_i q_i^{\beta p_i - tp_i + \alpha - 1}\, dq$$
$$= e^{-t\eta}\prod_i p_i^{tp_i}\frac{B(\beta p - tp_i + \alpha)}{B(\beta p + \alpha)}$$
$$= e^{-t\eta}\frac{\Gamma(\beta + d\alpha)}{\Gamma(\beta - t + d\alpha)}\prod_i p_i^{tp_i}\frac{\Gamma(\beta p_i - tp_i + \alpha)}{\Gamma(\beta p_i + \alpha)}. \tag{20}$$

Using the first-order Taylor approximation, we have the following estimates for log-gamma functions:

$$\log \Gamma(\beta + d\alpha) \leq \log \Gamma(\beta - t + d\alpha) + t\psi(\beta + d\alpha)$$
$$\log \Gamma(\beta p_i - tp_i + \alpha) \leq \log \Gamma(\beta p_i + d\alpha) - tp_i\psi(\beta p_i - tp_i + \alpha).$$

Inserting these inequalities and equation 19 into equation 20, we obtain

$$
\begin{aligned}
\Pr[D_{\mathrm{KL}}(p\|q) > \eta] &\leq e^{-t\eta} e^{t\psi(\beta + d\alpha)} \prod_i p_i^{tp_i} e^{-tp_i\psi(\beta p_i - tp_i + \alpha)} \\
&< e^{-t\eta} e^{t\log(\beta + d\alpha)} \prod_i p_i^{tp_i} e^{-tp_i \log(\beta p_i - tp_i + \alpha - 1/2)} \\
&= e^{-t\eta}(\beta + d\alpha)^t \prod_i p_i^{tp_i}(\beta p_i - tp_i + \alpha - 1/2)^{-tp_i} \\
&= e^{-t\eta}(\beta + d\alpha)^t \prod_i \left(\beta - t + p_i^{-1}(\alpha - 1/2)\right)^{-tp_i} \\
&= e^{-t\eta} \prod_i \left(\frac{\beta + d\alpha}{\beta - t + p_i^{-1}(\alpha - 1/2)}\right)^{tp_i} \\
&< e^{-t\eta} \prod_i \left(\frac{\beta + d\alpha}{\beta - t}\right)^{tp_i} \\
&= e^{-t\eta}\left(\frac{\beta + d\alpha}{\beta - t}\right)^t \\
&= \exp\left(-t\eta + t\log\frac{\beta + d\alpha}{\beta - t}\right) \\
&:= \exp(f(t)).
\end{aligned}
\tag{21}
$$

The function $f(t)$ is minimized at $t^* := \beta\left(1 - W\left(\frac{\beta e^{1+\eta}}{\beta + d\alpha}\right)^{-1}\right)$, where $W$ is the Lambert $W$ function. Note that $W$ satisfies the identity $\log(W(x)/x) = -W(x)$ for all $x \geq -e^{-1}$. Therefore,

$$
\begin{aligned}
f(t^*) &= -t^*\eta + t^* \log\frac{\beta + d\alpha}{\beta - t^*} \\
&= -t^*\eta + t^* \log\left\{\frac{\beta + d\alpha}{\beta})W\left(\frac{\beta e^{1+\eta}}{\beta + d\alpha}\right)\right\} \\
&= -t^*\eta + t^* \log\left\{\frac{\beta + d\alpha}{\beta e^{1+\eta}}W\left(\frac{\beta e^{1+\eta}}{\beta + d\alpha}\right)\right\} + t^* \log e^{1+\eta} \\
&= -t^*\eta - t^* W\left(\frac{\beta e^{1+\eta}}{\beta + d\alpha}\right) + t^*(1 + \eta) \\
&= t^*\left(1 - W\left(\frac{\beta e^{1+\eta}}{\beta + d\alpha}\right)\right) \\
&= -\beta\left(1 - W\left(\frac{\beta e^{1+\eta}}{\beta + d\alpha}\right)^{-1}\right)\left(W\left(\frac{\beta e^{1+\eta}}{\beta + d\alpha}\right) - 1\right).
\end{aligned}
\tag{22}
$$

The assumption $\beta \geq d\alpha/(e^{\eta/2} - 1)$ implies $\beta/(\beta + d\alpha) \geq e^{-\eta/2}$. We use the inequality $W(x) \geq \log x - \log\log x + \log\log x/(2\log x)$ for $x \geq e$ (Hoorfar & Hassani, 2008, Theorem 2.7) to obtain

$$W\left(\frac{\beta e^{1+\eta}}{\beta + d\alpha}\right) \geq W\left(e^{1+\eta/2}\right)$$

$$\geq 1 + \frac{\eta}{2} - \log\left(1 + \frac{\eta}{2}\right) + \frac{\log(1 + \eta/2)}{2(1 + \eta/2)}$$

$$= 1 + \frac{\eta}{2} - \left(\frac{1+\eta}{2+\eta}\right)\log\left(1 + \frac{\eta}{2}\right)$$

$$\geq 1 + \frac{\eta}{2} - \frac{\eta}{2} \cdot \frac{1+\eta}{2+\eta}$$

$$= 1 + \frac{\eta}{2(2+\eta)}.$$

Continuing from equation 22, we have

$$f(t^*) \leq -\beta\left(1 - \left(1 + \frac{\eta}{2(2+\eta)}\right)^{-1}\right)\left(1 + \frac{\eta}{2(2+\eta)} - 1\right) = -\beta\left(\frac{\eta^2}{2(2+\eta)(4+3\eta)}\right).$$

Inserting this inequality back into equation 21, we obtain

$$\Pr[D_{\mathrm{KL}}(p\|q) > \eta] \leq \exp(f(t)) \leq \exp(f(t^*)) \leq e^{-\beta\eta^2/(2(2+\eta)(4+3\eta))},$$

as desired.

