# OpenReview forum: "Dirichlet Mechanism for Differentially Private KL Divergence Minimization"
_TMLR — Accepted by TMLR_

### Review · Reviewer_a5Xg · 2022-12-19

**Summary Of Contributions:**

The paper considers the task of KL divergence minimization with differential privacy. They make the observation that KL divergence minimization with the exponential mechanism is almost equivalent to sampling from a Dirichlet distribution, with a mechanism they call the Dirichlet mechanism. They provide experimental results on Naive Bayes and Bayes networks.

**Audience:**

Yes

**Broader Impact Concerns:**

I think the broader impact section is adequate.

**Claims And Evidence:**

Yes

**Requested Changes:**

The only requested change that is critical to securing my recommendation is Weakness 2, and the following one:

It may be nice to plot the XE loss of random guessing as a horizontal line in Figure 3 (since label distribution and class imbalance vary between datasets). This would give a better sense for which epsilon values give meaningful results. This would also be nice in Figure 4.

The following would be nice-to-have things to address (in addition to weakness 1):

Theorem 1 proof is deferred to Appendix D, but it’s labeled Theorem 2 there. This would be nice to be consistent, and it would also be nice to restate theorems before they are proven in the Appendix.

Is it possible to have a lambda=1 result for Renyi DP?

It seems the privacy analysis requires r to be an exact root, is there any tolerance for numerical instabilities or other reasons a root-finding algorithm may return imprecise results?

**Strengths And Weaknesses:**

Strengths:
The main calculation made by the paper to convert the exponential mechanism to the Dirichlet mechanism is nice. The Dirichlet distribution is easy to sample from, and KL divergence is well-used, so this is a nice derivation.

The cross entropy loss for all the experiments is pretty promising.

Weaknesses:
The cross entropy loss improvements don’t seem to translate to improved accuracy for the Naive Bayes models. It’s unclear exactly why this is, and there’s not any attempt to explain this.

The Dirichlet mechanism does not permit DP with delta=0, but the exponential mechanism is a mechanism which guarantees delta=0. It’s pretty unclear from the analysis in Section 2.2 where this gap shows up. It looks to me like this can be from the beta term, which should be the privacy parameter, being data dependent. This should be better explained, but fortunately the proof of Theorem 1 doesn’t seem to rely on this.

---

> ### Author Response · Authors · 2022-12-22
> **Response to Reviewer a5Xg**
>
> Thank you for the comments. We will address the reviewer's concerns below, and we will upload a revision after all 3 reviews are available.
>
> > The cross entropy loss improvements don’t seem to translate to improved accuracy for the Naive Bayes models.
>
> This is because the cross entropy (XE) loss is a continuous function of the predicted probability, while the accuracy is a result of thresholding the probability. For example, suppose that the true label of an input $x$ is $1$. Two predicted probabilities $\Pr(y=1\mid x)=0.1$ and $\Pr(y=1\mid x)=0.4$ are not distinguishable when computing the accuracy, but the XE losses of the former is much smaller than the latter. Suppose that there are two models with similar accuracies, the one with smaller XE loss tends to not be over-confident in their incorrect predictions. For example, if the true label of $x$ is $1$, then $\Pr(y=0\mid x)$ is much less than $1$.
>
> > The Dirichlet mechanism does not permit DP with delta=0, but the exponential mechanism is a mechanism which guarantees delta=0. It’s pretty unclear from the analysis in Section 2.2 where this gap shows up.
>
> The DP result for the exponential mechanism (with finite $\varepsilon$) requires that the change in the loss function upon moving to a neighboring dataset must be uniformly bounded above. In other words, $\Delta D_{KL} \coloneqq \sup_{x,x',y}\lvert D_{KL} (\widetilde{f(x)}\Vert y)-D_{KL} (\widetilde{f(x')}\Vert y) \rvert$, where the supremum is taken over all neighboring $x$ and $x'$, must be finite. However, assuming that $\widetilde{f_i(x)}\not= \widetilde{f_i(x)}$ for some $i$, we can choose $y_i\to 0$ so that $\Delta D_{KL}\to \infty$. Thus the usual DP result for the exponential mechanism cannot be applied to the Dirichlet mechanism here.
>
> > It may be nice to plot the XE loss of random guessing as a horizontal line in Figure 3 (since label distribution and class imbalance vary between datasets). This would give a better sense for which epsilon values give meaningful results. This would also be nice in Figure 4.
>
> We have added plots of XE losses of random guessing, which is a $0$-DP model, as suggested. The plot can be viewed **[Here](https://picallow.com/wp-content/uploads/2022/12/2022-12-22_63a495fe61f1e_XE.png)**.
> On some datasets, the XE loss of the random guessing is very low compared to the private models. This is because private models tend to be over-confident in their incorrect predictions on some data points, which cause the high XE losses. The random-guessing model, on the other hand, does not suffer from this problem. Nonetheless, the random-guessing model has very low accuracy by default, while the Dirichlet mechanism, which tries to minimize the KL divergence between its private parameters and the maximum likelihood estimators (MLE), will inherit some accuracy from the MLE (as illustrated in Figure 4).
>
> We decide not to add the plots of the random-guessing model to Figure 4 as the accuracies are too low ($\approx 0.5$ in the binary classification).
>
> > Theorem 1 proof is deferred to Appendix D, but it’s labeled Theorem 2 there. This would be nice to be consistent, and it would also be nice to restate theorems before they are proven in the Appendix.
>
> The label of Appendix D should be Theorem **1** here.
>
> > Is it possible to have a lambda=1 result for Renyi DP?
>
> Theorem 1 holds for $\lambda =1$ as well. Taking the limit $\lambda\to 1$, the Renyi divergence becomes the KL divergence, and there is a formula for the KL divergence between two Dirichlet distributions $p\sim\text{Dirichlet}(u)$ and $q\sim\text{Dirichlet}(u')$:
>
> $$
> D_{KL} (p\Vert q) =\log \Gamma(u_0)-\sum_{i} \log \Gamma(u_i)-\log \Gamma(u'_0)+\sum_i \log \Gamma(u'_i)+\sum_i (u_i-u'_i) (\psi(u_i)-\psi(u_0)),
> $$
>
> where $u_0=\sum_i u_i$ and $u'_0=\sum_i u'_i$. Let $u=rf(x)+1$, $u'=rf(x')+1$. We split the right-hand side into two parts and apply the Taylor approximation:
>
> $$
> -\sum_{i} \log \Gamma(u_i)+\sum_i \log \Gamma(u'_i)+\sum_i (u_i-u'_i) \psi(u_i) \leq \frac1{2} \sum_i (u_i-u'_i)^2 \psi'(1),
> $$
>
> $$
> \log \Gamma(u_0)-\log \Gamma(u'_0)-\sum_i (u_0-u'_0) \psi(u_0) \leq -\frac1{2} \sum_i (u_i-u'_i)^2 \psi'(\max\\{u_0,u'_0\\}) \leq 0.
> $$
>
> Adding these two inequalities yields Theorem 1 with $\lambda=1$.
>
> > It seems the privacy analysis requires r to be an exact root, is there any tolerance for numerical instabilities or other reasons a root-finding algorithm may return imprecise results?
>
> It is proved in Lemma 2 (Appendix C) that the function $r \mapsto f(r) = \frac1{2}\lambda r^2 \Delta_2^2\psi'(1 + 3(\lambda − 1)r\Delta_{\infty})$ is continuous and strictly increasing from $0$ to $\infty$. Thus any off-the-shelf root-finding algorithm on $f$ should be numerically stable. If $\tau$ is the tolerance, we just need to make sure that $\varepsilon-\tau<f(r)<\varepsilon$ in order to guarantee $(\lambda,\varepsilon)$-RDP.
>
> Let us know if there are any further questions or comments.

---

> > ### Comment · Reviewer_a5Xg · 2023-01-30
> > **I appreciate the reply**
> >
> > Thank you for your reply. I think the random guessing cross entropy line is very helpful to help contextualize XE improvements. I might also suggest shrinking the y scale for these plots where random and nonprivate are very close to each other, since the "interesting range" for loss values here is between these two. This is just a suggestion, and I appreciate the authors' efforts in improving the paper. I have no further comments on other questions I raised.

---

> > > ### Author Response · Authors · 2023-01-30
> > > **Thank you for the review**
> > >
> > > We thank the reviewer for the suggestion and kind words. We have modified the plots of the cross-entropy and the log-likelihood to be in logarithmic scale to accentuate the gaps between the Dirichlet mechanism and the non-private algorithm.

---

### Review · Reviewer_XPY8 · 2022-12-31

**Summary Of Contributions:**

The paper discusses a Dirichlet mechanism for reporting a distribution computed on sensitive data, and the goal is for the privately reported distribution to be close to the true distribution in KL divergence. The solution is an application of the exponential mechanism and is shown to satisfy RDP.

**Audience:**

No

**Claims And Evidence:**

Yes

**Requested Changes:**

I feel that the paper needs significant improvements before publication, most notably a detailed comparison (in theory and experiments) to prior work on private reporting of histograms/distributions. When is the proposed mechanism better and in what sense? What is the best possible existing baseline? What is the comparison to Gohari et al.? How does the comparison change as you vary both epsilon and lambda?

**Strengths And Weaknesses:**

The solution is simple and intuitive, but to me it's not clear what this paper is trying to accomplish in the context of existing work. The proposed solution is essentially a mechanism for reporting histograms, and this is a well-studied problem in the literature on differential privacy. This paper seems to want to report (normalized) histograms with the goal of achieving good utility in terms of the KL divergence specifically. Why is this so important?

Nevertheless, let us assume that we care about reporting histograms with good KL utility. Then, I still think that there is not enough comparison to existing baselines. In the related work section, you mention the work of Gohari et al. What exactly are the similarities and differences between your work and this prior work? A thorough comparison is needed. It sounds like you're saying that the work of Gohari et al. involves some complications that your work avoids, but your mechanism is also arguably more complicated than the classical Laplace mechanism baseline. In the experimental section, you compare to the Gaussian and Laplace mechanism. How exactly did you arrive at the noise scaling for these mechanisms? The noise scaling looks loose. The experiments also confirm that the utility of the paper really lies in prioritizing KL divergence, because the proposed mechanism doesn't seem to fare well in terms of accuracy in Figure 4. Finally, you fix lambda = 5 throughout; I think you should show the comparison across different values of lambda.

Minor:
- At the botom of page 1, you have an example where you imagine a scenario that happens with probability 0 (noise is [-0.1,0.1]). That illustration doesn't sound convincing to me.
- In the notation section, you say things like "For any positive reals x and x′, the notation x \propto x′ means x = Cx′ for some constant C > 0". This doesn't compile. You're talking about functions/sequences, not positive real numbers.
- At the top of page 4, you study the bias of y_i. It would also be interesting to look at the variance, because the variance should grow as alpha gets small.
- At the beginning of the conclusion section, you have a typo ("We study derive ...").

---

> ### Author Response · Authors · 2023-01-27
> **Response to Reviewer XPY8**
>
> Thank you for the comments. We have revised the paper to address your concerns; the changes are highlighted in yellow. Let us further clarify some points here.
>
> > ...The proposed solution is essentially a mechanism for reporting histograms...
>
> Reporting histograms is not the main application of our mechanism. In fact, the Dirichlet mechanism is not suitable for this task as the performance of the report is usually measured in $\ell^1$ or $\ell^2$ distance, but the Dirichlet mechanism is designed to minimize the KL divergence, which arises naturally in probabilistic and Bayesian modeling: fitting a probabilistic models is typically done with log-likelihood maximization, a Bayesian model is usually evaulated using Akaike information criterion or Bayesian information criterion. Recently, it is a common practice to use out-of-sample log-likelihood to evaluate a Bayesian model [1]. All of these tasks can be cast as KL divergence minimization, and so the Dirichlet mechanism would help performing these tasks under privacy constraints.
>
> Thus, the Dirichlet mechanism is designed with the goal of reporting the parameters of probabilistic models in mind, not publishing a single histogram.
>
>
> > ...The noise scaling looks loose...
>
> Thank you for pointing this out. The scaling for the Gaussian mechanism is already correct; we have an $\ell^2$-sensitivity of $\Delta^2_2=\sqrt{2}$, so, according to [2], the variance for each of $K$ Gaussian mechanism is $K\lambda\Delta^2_2/(2\varepsilon)=K\lambda/\varepsilon$.
>
> The scaling for the Laplace mechanism, as you pointed out, is not correct. We have replace the scaling with the one derived in [2]. We have looked at several papers published in 2022, and people are still using the same scaling.
>
>
> < ...It would also be interesting to look at the variance, because the variance should grow as alpha gets small.
>
> We have added a variance analysis of a Dirichlet variable in Section 2.2. You can see it below the bias analysis.
>
>
> > ...I feel that the paper needs significant improvements before publication, most notably a detailed comparison (in theory and experiments) to prior work on private reporting of histograms/distributions. When is the proposed mechanism better and in what sense? What is the best possible existing baseline? What is the comparison to Gohari et al.? How does the comparison change as you vary both epsilon and lambda?...
>
> All prior work studied private reporting of histograms with the goal of minimizing the $\ell_1$- or $\ell^2$-distance, for which the gold-standard method is the Laplace mechanism and Gaussian mechanism, respectively.
>
> We are the first to study private KL minimization (not for reporting a single histogram but for reporting parameters of probabilistic models), and the best possible baseline for our mechanism right now is the Dirichlet mechanism proposed by Gohari et al.
>
> [1] Aki Vehtari, Andrew Gelman, and Jonah Gabry. Practical bayesian model evaluation using leave-one-out
> cross-validation and WAIC. Statistics and Computing, 27(5):1413–1432.
>
> [2] Ilya Mironov. Rényi differential privacy. In 30th IEEE Computer Security Foundations Symposium, CSF 2017, pp. 263–275.

---

### Review · Reviewer_5JaD · 2023-01-18

**Summary Of Contributions:**

The authors propose a version of the Dirichet mechanism that satisfies Renyi differential privacy. This algorithm privately outputs a distribution that is close in KL divergence to an empirical distribution. This makes it useful for naïve bayes classification and maximum likelihood estimation of Bayesian networks.

While this algorithm looks like an application of the standard exponential mechanism from differential privacy (a non-obvious connection the authors establish), the privacy and utility statements are not covered by the standard results. The authors give privacy and utility bounds. The proof of privacy bounds in particular are different to the standard privacy proofs for the exponential mechanism in a non-trivial way.

RDP guarantees for the Dirichlet mechanism have been studied in prior work, but the algorithm presented in this work is much more practical and efficient to implement.


**Audience:**

Yes

**Claims And Evidence:**

Yes

**Requested Changes:**

- Improve the prose, providing more background for the reader. Specific examples given in review above.
- Discuss comparison with Gohari et al. (2021).

**Strengths And Weaknesses:**

- An efficient tight privacy analysis for the Dirichlet mechanism is a main contribution of this paper. The authors demonstrate the tightness of their bound by comparing it to a numerical computation of the Renyi divergence between the mechanism run on two neighboring databases (Figure 2). This figure is pretty convincing in highlighting the tightness of the result.
- The experiments the authors perform show that the proposed algorithm outperforms Gaussian and Laplacian perturbation algorithms. These seem like a reasonable baseline, although it would be interesting to know if there are more reasonable algorithms to compare to (e.g. the exponential mechanism with a truncated KL divergence?)
- Having the motivation for KL divergence minimization fleshed out more would have been helpful. The description of the applications/motivating examples was short. In particular, I am not very familiar with the “test CE Loss” vs. “test accuracy” discussion. The proposed algorithm is designed to minimize the test CE loss, and performs well at this task, but does comparably to the more naive algorithms for the test accuracy. I assume this is something inherent about these two metrics (not the algorithm) but further discussion would help motivate prioritizing the test CE loss. Figure 4 (bank) was particular confusing.
- Although the algorithm proposed fits into the structure of the exponential mechanism, due to the infinite sensitivity of the KL divergence, it is not an instance of the exponential mechanism. Therefore, the privacy and utility proofs are not covered by the standard theorems. This is mentioned in the introduction but I thought further discussion was warranted to motivate the problem. For example, the proofs are quite different to those given in the standard theorems, and this seems worth pointing out. Similarly, further discussion of Geumlek et al. (2017) and Gohari et al. (2021) would help motivate the problem. In particular, a numerical analysis comparing to Gohari et al. would be useful to assess how this work improves over that work.
- Although the results are interesting, the presentation of the results could use a lot of improvement. I thought that throughout the paper, the authors required a lot of the reader, skipping connecting sentences and background information that would greatly improve readability. I often found myself having to fill in information.
- I did not find any errors in the proofs of the theorems.

Minor Comments:
- Places where more information would be useful. In many of these places just an extra sentence or two guiding the reader would substantially improve readability.
    - The introduction of the problem at the bottom of page 1 needed more explanation. A simple statement stating that we were going to interpret the the count data as discrete pdfs would improve readability.
    - Terms are often used without explanation (e.g. polygamma). These are standard terms, but it is helpful to include a definition since they form an important part of the theorem and proofs.
    - I understand brevity is a priority in discussing related work but, without prior knowledge I would not have been able to follow some of the related work. E.g. when discussing Wang et al. 2015, the mechanism is only described as “probability sampling”, what is this?
    - I found the discussion of the proof of Theorem 2 in the main body (pg 6-7) quite confusing. In particular, it was not clear to me why eps>=1 implied r>1 and phi’(1+2(lamb-1)rDelta)=Theta(1/(lambda-1)r)? Was this explained?
    - In the applications section (5.1), more explanation would be helpful. The authors go straight into technical material and a reader unfamiliar with naive Bayes models would struggle to follow.
    - A final line in the proof of Lemma 2 stating why the fact that phi(x) is increasing and limits towards 0 and infinity implies a unique solution for every eps would help the reader.
- The third paragraph of the introduction oversimplified the vast research on differential privacy, claiming that many designs are simply output perturbation using Laplacian or Gaussian noise. The authors are aware of the exponential mechanism, which does not fall in this category.
- On page 3, “In contrast to pure and approximate DP,…”. I don’t think “in contrast” is the right phrase here.
- When introducing the exponential mechanism “beta is a privacy-related parameter” is a little vague, “function of epsilon” would be more clear.
- On page 4 when discussing the bias, it should be clear that this bias is only over the randomness in the algorithm, not the population.
- Typos:
    - pg. 8 “in terms of accuracy, there are no clear winner”
    - Pg. 10 “We study define”
    - Mislabelled equation in the proof of Lemma 3?
    - Psi’ is deceasing (not increasing) in appendix D.
    - At end of appendix D, after “Therefore, continuing on from equation 12,”, x_i-x_i’ should be f_i(x)-f_i(x’).
- I don’t think the “parent” relationship in a Bayesian network was ever defined.
- I don’t think B or alpha_m were defined in appendix D

---

> ### Author Response · Authors · 2023-01-27
> **Response to Reviewer 5JaD**
>
> Thank you for very detailed comments, and for giving this paper a thorough read. We have revised the paper to address your concerns; the changes are highlighted in yellow.  Let us further clarify some points here.
>
> Strengths And Weaknesses:
>
> > ...it would be interesting to know if there are more reasonable algorithms to compare to (e.g. the exponential mechanism with a truncated KL divergence?)...
>
> We have tried the exponential mechanism with a truncated KL divergence via rejection sampling, but even when $\varepsilon=10$, it would take for ever to accept a sample.
>
> > further discussion would help motivate prioritizing the test CE loss. Figure 4 (bank) was particular confusing.
>
> We have added more discussion on the test CE loss versus the accuracy in the experiment section.
>
>
> > ...further discussion of Geumlek et al. (2017) and Gohari et al. (2021) would help motivate the problem. In particular, a numerical analysis comparing to Gohari et al. would be useful to assess how this work improves over that work.
>
> We have added some numerical comparisons between our guarantee and Gohari et al.'s guarantee. We think it would be impossible to compare the guarantees analytically as Gohari et al.'s results are quite complicated (we have add more discussion in the "Related work" section). Looking at their proof, they obtain a guarantee for $\varepsilon$ by squashing all Dirichlet parameters to only two beta parameters. Our work should generally provide a better guarantee as we keep all the parameters and instead use the Taylor approximation.
>
> > ...it was not clear to me why eps>=1 implied r>1 and phi’(1+2(lamb-1)rDelta)=Theta(1/(lambda-1)r)...
>
> Sorry for the confusion, we meant to say $r=\omega(1)$. For the second statement, we use $\psi'(x) = \Theta(1/x)$ for $x > 1$.
>
>
> > Requested Change: Improve the prose, providing more background for the reader. Specific examples given in review above.
>
> Thank you for the detailed comments and clear examples. We have followed your suggestions and made several changes in the paper.
>
> > Requested Change: Discuss comparison with Gohari et al. (2021).
>
> > We have added some discussion comparing our method to Gohari et al. (2021) in the "Related work" section. We have also added more experiments comparing between these two methods in the experiment section and the Appendix C.

---

> > ### Comment · Reviewer_5JaD · 2023-01-27
> > **revisions**
> >
> > Thank you for the implemented revisions and including the comparison to Gohari et al. I believe the changes have greatly improved readability.
> >
> > A minor note: reviewer XPY8’s comment about the notation x \prop x’ does not seem to be fixed. This notation still does not make sense when x and x’ are scalars, they should be sequences/functions for this notation to compute.

---

> > > ### Author Response · Authors · 2023-01-27
> > > **Fixed the definition of $\propto$ to be defined for functions**
> > >
> > > Thank you for the quick response and for the correction. The definition of $f(x) \propto f'(x)$ should now be correct.

---

### Comment · Action_Editors · 2023-01-19
**Revision and discussion**

Thanks to all reviewers for getting their reviews in. Discussion and revision period now begins. It seems like the primary (but not only) concerns are related to writing and comparison with related work. The authors are recommended to make these changes to the draft and submit it as a revision, for the reviewers to inspect.

The reviewers will be able to submit decisions within about two weeks from now, so the authors may want to keep that timeline in mind.

---

### Author Response · Authors · 2023-01-27
**Official response from the authors**

We thank the reviewers for helpful and detailed comments. We have made the following changes to the paper. The changes are highlighted in yellow:

- We have added more discussion of the Gohari et al. (2021)'s method in the "Related work" section, along with a simulation to illustrate why their method can be numerically unstable.

- We have added a series representation of the polygamma function $\psi'$, showing that we can efficiently approximate the function at any precision.

- We have added a simulation comparing our privacy guarantee (in terms of approximate DP) and Gohari et al.'s privacy guarantee.

- We have performed more experiments on the Naive Bayes and Bayesian networks, comparing several models (including Gohari et al.'s) satisfying a range of approximate DP guarantees (instead of the RDP guarantees). The results of the experiments are in Appendix C.

- We have added the proof of Theorem 1 for the case $\lambda = 1$.

- We have added a short analysis of the effects of $r$ and $\alpha$ on the variance of Dirichlet sample in Section 2.2.

- We have clarified in the Introduction that the Dirichlet mechanism is designed with the goal of private probabilistic/Bayesian modeling in mind, and not reporting a single histogram

- We have added more discussion on the cross-entropy loss versus the accuracy in the first experiment.

- We have improved the prose and given more background for the reader throughout the paper.

Thank you again for all the comments, which help improve the paper by a lot. If there are any further points for us to clarify, please let us know.

---

### Decision · Action_Editors · 2023-02-20

**Recommendation:** Accept with minor revision

**Comment:**

Overall, the reviewers felt this was a solid and sound piece of work. The only remaining critique was from XPY8, who was unclear about the application/motivation of the given work, since it focuses on estimating the empirical distribution, rather than the population itself. The authors replied to this, sketching an argument about how it may imply things for the population setting (https://openreview.net/forum?id=lmr2WwlaFc&noteId=diapYqyugI). The authors are requested to add discussion and clarification to the paper on the distinction between the empirical and population settings, and whether results hold for which settings. For more on the population setting, some references the authors could consult include https://proceedings.mlr.press/v40/Kamath15.html and https://arxiv.org/abs/2302.06869.

**Audience:**

The audience for this paper is the differential privacy community, which is well-represented in the TMLR audience.

**Claims And Evidence:**

This paper proposes the Dirichlet mechanism, along with proofs of (differential) privacy and utility. Empirical evaluation is provided as well. The proofs appear sound, and the empirical evaluation is thorough and convincing.

---

> ### Author Response · Authors · 2023-02-21
> **Reply to the Action Editors**
>
> Thank you for the recommendation and the references. We have clarified in Section 2.2 that this study only focuses on estimation of an empirical distribution. We have also modified a couple of sentences in the introduction an the conclusion to emphasize this point as well. The changes made are highlighted in yellow.

---

> > ### Comment · Action_Editors · 2023-02-21
> > **Thanks**
> >
> > I personally would have liked to see some comment on what you can actually do in this setting. In particular, your comment linked above seems to sketch that there are some implications for distribution estimation. But in the final paper, you are claiming that it is for future study.

---

> > > ### Author Response · Authors · 2023-02-21
> > > **Reply #2**
> > >
> > > Thank you. We have added the discussion for distribution estimation as another application of the Dirichlet mechanism in Section 2.2. Please let us know if you there is still anything unclear.

---

> > > > ### Comment · Action_Editors · 2023-02-21
> > > > **Cool**
> > > >
> > > > Generally seems good. Minor comments:
> > > > - I appreciate the highlighting for taking a look at the difference, but please remove it (and de-anonymize) for the final version.
> > > > - In (4), is there a reason you converted the - to + in the second term? Since KL is non-negative, can't you simply drop this term to get a valid upper bound?
> > > > - This bound could be quite bad in some cases, where the probability of some symbols is extremely small and the KL is in a certain direction, requiring a very large number of samples for the empirical estimator to converge in these cases. Thus, deriving finite sample bounds is an interesting question for future investigation. You may want to note some of these points to encourage follow-up work.

---

> > > > > ### Author Response · Authors · 2023-02-21
> > > > > **Thank you**
> > > > >
> > > > > Thank you for the comments. You are right; we can actually disregard the negative KL divergence term. I do believe that to find a finite sample bound, keeping this term (with the negative sign) might be helpful.
> > > > >
> > > > > I have addressed the issue of this bound that requires further investigation. And I have deanonymized the paper.

---

> > > > > > ### Comment · Action_Editors · 2023-02-21
> > > > > > **Almost there**
> > > > > >
> > > > > > Awesome, can you add 02/2023 instead of MM/YYYY? Then I think we're good to go.

---

> > > > > > > ### Comment · Action_Editors · 2023-02-21
> > > > > > > **Oh and...**
> > > > > > >
> > > > > > > ...the official OpenReview link instead of XXXX

---

> > > > > > > > ### Author Response · Authors · 2023-02-21
> > > > > > > > **Done**
> > > > > > > >
> > > > > > > > Sorry about that. The month/year and the OpenReview link have been added.